# Neuron-Level Analysis of Cultural Understanding in Large Language Models

**Taisei Yamamoto**
The University of Tokyo, Riken
yamamo96@is.s.u-tokyo.ac.jp

**Ryoma Kumon**
The University of Tokyo, Riken
kumoryo9@is.s.u-tokyo.ac.jp

**Danushka Bollegala**
University of Liverpool, Amazon
danushka@liverpool.ac.uk

**Hitomi Yanaka**
The University of Tokyo, Riken, Tohoku University
hyanaka@is.s.u-tokyo.ac.jp

## Abstract

As large language models (LLMs) are increasingly deployed worldwide, ensuring their fair and comprehensive cultural understanding is important. However, LLMs exhibit cultural bias and limited awareness of underrepresented cultures, while the mechanisms underlying their cultural understanding remain underexplored. To fill this gap, we conduct a neuron-level analysis to identify neurons that drive cultural behavior, introducing a gradient-based scoring method with additional filtering for precise refinement. We identify *culture-general* neurons contributing to cultural understanding regardless of cultures, and *culture-specific* neurons tied to an individual culture. *Culture-general* and *culture-specific* neurons account for less than 1% of all neurons and are concentrated in shallow to middle MLP layers. We validate their role by showing that suppressing them substantially degrades performance on cultural benchmarks (by up to 30%), while performance on general natural language understanding (NLU) benchmarks remains largely unaffected. Moreover, we show that *culture-specific* neurons support knowledge of not only the target culture, but also related cultures. Finally, we demonstrate that training on NLU benchmarks can diminish models' cultural understanding when we update modules containing many *culture-general* neurons. These findings provide insights into the internal mechanisms of LLMs and offer practical guidance for model training and engineering. Our code is available at https://github.com/ynklab/CULNIG

## 1 Introduction

LLMs are rapidly spreading throughout the world with their ability to solve various tasks. Our world is culturally diverse, and our knowledge, commonsense, and values are not always universal. LLMs must possess cultural understanding to be deployed fairly and prevent cultural inequity. However, several studies have pointed out that LLMs, which are mainly trained on English-dominant corpora, often exhibit culture-related biases, generating outputs skewed toward certain highly represented cultures (Naous et al., 2024; Myung et al., 2024; Sukiennik et al., 2025). In order to evaluate the cultural understanding of LLMs, a number of benchmarks have been constructed (Myung et al., 2024; Chiu et al., 2025; Rao et al., 2025; Zhao et al., 2024, *inter alia*). Additionally, some methods have been proposed to enhance cultural awareness of LLMs (Li et al., 2024a;b; Liu et al., 2025). Nonetheless, the mechanisms behind the cultural understanding of LLMs have not been well investigated. In order to improve the cultural understanding of LLMs efficiently and robustly, it is desirable to elucidate the inner workings by which LLMs perform culture-related inference.

Previous studies have applied neuron-level analysis to investigate various properties of LLMs, such as social bias (Yang et al., 2024) and personality (Deng et al., 2025). As for cultural mechanisms, Ying et al. (2025) analyzed neurons activated most strongly when the prompt language aligns with the cultural content. In addition, Namazifard & Galke (2025) proposed a method to disentangle culture neurons from language neurons. These studies primarily examine culture in relation to lan-

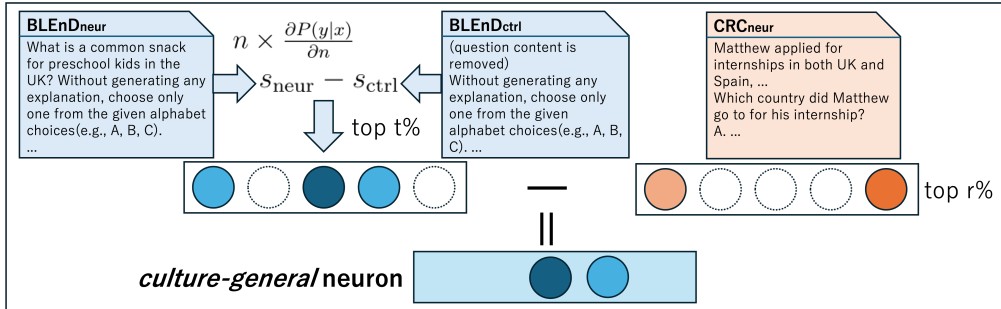

Figure 1: An overview of CULNIG when identifying *culture-general* neurons. We first select the top $t\%$ of the neurons ranked by gradient-based attribution scores on $\text{BLEnD}_{\text{neur}} - \text{BLEnD}_{\text{ctrl}}$ ($s_{\text{neur}} - s_{\text{ctrl}}$) to find neurons contributing to cultural mechanisms. By subtracting $s_{\text{ctrl}}$, we exclude neurons facilitating task understanding. We then remove the top $r\%$ of the neurons on $\text{CRC}_{\text{neur}}$ to filter out superficial neurons activated by country names.

guage, rather than how LLMs shape their behavior based on cultural information. Moreover, they rely on activation-based methods, which can be imprecise because cultural representations are not necessarily encoded in every token of culturally relevant texts.

In this paper, we explore three research questions: (i) the existence and distribution of *culture-general* neurons that contribute to cultural understanding across cultures, (ii) the differences of *culture-specific* neurons across cultures and the correlation between these neurons and cultural relations, and (iii) the potential engineering applications of our neuron analysis. We interpret cultural understanding along two dimensions: (a) knowledge that is specific to a particular culture and (b) the ability to capture differences in values across diverse cultural backgrounds. To address these questions, we introduce **CUL**ture **N**euron **I**dentification Pipeline with **G**radient-based Scoring (**CULNIG**, Figure 1), a method to accurately identify neurons that contribute to the cultural understanding of LLMs. CULNIG employs gradient-based attribution scores to quantify the contribution of each neuron to the outputs, and a control dataset to exclude neurons associated with task understanding rather than cultural understanding. We also construct the CountryRC (Country Reading Comprehension, CRC) dataset to filter out superficial neurons that are irrelevant to cultural understanding.

We comprehensively evaluate the identified neurons using cultural benchmarks and general natural language understanding (NLU) benchmarks that do not necessarily require cultural understanding. We find that masking *culture-general* neurons significantly degrades the cultural understanding of LLMs while having only minor impacts on the NLU tasks. Importantly, although CULNIG covers only a subset of cultural knowledge categories to identify neurons, *culture-general* neurons generalize to broader cultural mechanisms, encompassing different knowledge domains, cultural values, and multilingual settings. *Culture-general* neurons account for fewer than 1% of all neurons and are concentrated in the MLP modules of shallow to middle layers. We further show that masking *culture-specific* neurons leads to LLMs losing cultural knowledge of the target and related cultures. Moreover, we demonstrate that when we fine-tune a model with NLU datasets, updating modules containing many *culture-general* neurons can cause greater degradation of cultural understanding after training. These findings illustrate how insights into the inner workings of LLMs can inform practical engineering decisions.

## 2 RELATED WORK

### 2.1 EVALUATING CULTURAL UNDERSTANDING OF LLMS

Several cultural benchmarks have been developed to measure the cultural understanding of LLMs. BLEnD (Myung et al., 2024) covers everyday knowledge across 16 cultures in six categories, with multilingual short answer questions and English multiple-choice questions (MCQs). CulturalBench (Chiu et al., 2025) is an MCQ benchmark of cultural knowledge spanning 45 countries. NormAd (Rao et al., 2025) evaluates cultural etiquette through daily-life scenarios, asking whether

the behaviors are acceptable in the target country. WorldValuesBench (Zhao et al., 2024), derived from World Values Survey (WVS) Wave 7 (Haerpfer et al., 2020), assesses understanding of cultural values by a prediction task of survey responses based on demographic attributes.

Prior studies have pointed out that LLMs often exhibit cultural biases toward highly represented cultures in training corpora (Naous et al., 2024; Myung et al., 2024; Sukiennik et al., 2025). Ying et al. (2025) demonstrates Cultural-Linguistic Synergy, a phenomenon where the performance of LLMs on cultural benchmarks improves when the prompt language agrees with the cultural content. In contrast, Myung et al. (2024) reports that Cultural-Linguistic Synergy does not always appear for low-resource languages, where limited language proficiency may act as a bottleneck. Building on these studies, we analyze the cultural understanding and behavior of LLMs at the neuron level, utilizing existing cultural benchmarks.

## 2.2 Neuron-Based Interpretability Analysis

Mechanistic interpretability attempts to uncover the internal mechanisms of LLMs, with many studies focusing on neurons as the unit of analysis. Dai et al. (2022) proposed a gradient-based attribution method to identify neurons that express a certain knowledge. They show that only a few knowledge neurons in deep layers support factual recall in BERT (Devlin et al., 2019). Using similar gradient-based attribution, Chen et al. (2025) located query-relevant neurons that facilitate question answering, and Yang et al. (2024) found bias neurons and mitigated bias by pruning them.

Several methods employ the activation probability to identify neurons. Tang et al. (2024) and Kojima et al. (2024) identified language-specific neurons that are activated when LLMs are prompted in a specific language, with the former introducing LAPE (Language Activation Probability Entropy). Ying et al. (2025) analyzed neurons underlying Culture-Linguistic Synergy. Namazifard & Galke (2025) proposed CAPE (Culture Activation Probability Entropy) to isolate culture neurons from language-specific neurons of LAPE, using a dataset of culturally diverse texts.

However, these methods often lack comprehensive evaluation across multiple cultural understanding benchmarks. Moreover, although both positive and negative activations encode useful information, activation-based approaches consider only positive activations while clipping negative activations to zero activation probability. Also, cultural content is not necessarily expressed in every token, unlike languages. Thus, activation probabilities may not be suitable for identifying culture neurons. Therefore, we adopt a gradient-based attribution approach and validate the identified neurons across multiple benchmarks spanning different cultural attributes.

## 3 Methods

In this section, we introduce CULNIG to identify *culture-general* and *culture-specific* neurons that directly support cultural understanding. Removing these neurons is expected to substantially alter model behavior on cultural benchmarks, unlike neurons that merely respond to culture-related tokens.

## 3.1 Neurons in LLMs

Each layer of a neural network can be represented as a hidden vector whose dimensions correspond to neurons. We refer to Yu & Ananiadou (2024) for the concept of neurons in LLMs. Let $h_i^{(l)}$ denote the hidden vector of the $i$-th token at the $l$-th layer. In transformer-based LLMs (Vaswani et al., 2017), the $l$-th layer transforms its input as $h_i^{(l)} = h_i^{(l-1)} + a_i^{(l)} + f_i^{(l)}$, where $a_i^{(l)}$ and $f_i^{(l)}$ denote the outputs of the attention and MLP modules, respectively. For MLP modules, recent LLMs commonly employ gated linear units (GLUs) (Shazeer, 2020), which consists of gate, up, and down projections. Let $W_{\text{gate}}^{(l)}, W_{\text{up}}^{(l)} \in \mathbb{R}^{N \times d}, W_{\text{down}}^{(l)} \in \mathbb{R}^{d \times N}$ denote the corresponding projection matrices, where $d$ is the hidden size and $N$ is the intermediate size.

Geva et al. (2021) show that MLP modules can be interpreted as key-value memories. The output $f_i^{(l)}$ is expressed as a weighted sum of the column vectors of $W_{\text{down}}^{(l)}$ (subvalues), and the weights are computed from the neurons of gate and up projections, that is, the inner products of the inputs and

the row vectors of $\boldsymbol{W}_{\text{gate}}^{(l)}$ and $\boldsymbol{W}_{\text{up}}^{(l)}$ (subkeys). By analyzing the contribution of these intermediate neurons, MLP outputs can be decomposed into a sum of subvalues. For MLPs, we focus on neurons in the gate projection, since the gate and up projections share the same subvalue, and gate neurons play a role as a gate to determine whether they pass the weights. Similarly, the output of an attention module $\boldsymbol{a}_i^{(l)}$ can be decomposed into a weighted sum of the column vectors of an output matrix $\boldsymbol{W}_o^{(l,h)} \in \mathbb{R}^{d \times D}$, where $D$ is the head dimension, and the weights are determined by query, key, and value vectors. Thus, we investigate neurons from the query, key, and value modules.

## 3.2 Neuron Attribution Scores

In order to quantify the importance of each neuron on a given instance, we adopt the method based on Yang et al. (2024). Let $P(y|x)$ denote the probability of the output sequence $y$ assigned by the model when given an input sequence $x$. The attribution score of the $l$-th layer $k$-th neuron $n^{(l,k,i)}$ at the $i$-th token position is calculated using the following formula:

$$s^{(l,k,i)}(x,y) = n^{(l,k,i)} \times \frac{\partial P(y|x)}{\partial n^{(l,k,i)}} \tag{1}$$

We then take the maximum score across token positions:

$$s^{(l,k)}(x,y) = \max_i s^{(l,k,i)}(x,y) \tag{2}$$

Note that Equation 1 can be viewed as a first-order approximation of the causal effect of neuron $n^{(l,k,i)}$. Let $P(y|x, n^{(l,k,i)} = u)$ denote the output probability when the activation value of $n^{(l,k,i)}$ is $u$, and $\bar{u}$ denote the actual activation. The causal effect of $n^{(l,k,i)}$ on probability is $P(y|x, n^{(l,k,i)} = \bar{u}) - P(y|x, n^{(l,k,i)} = 0)$. Here, we expand $P(y|x, n^{(l,k,i)} = u)$ around $\bar{u}$ using the Taylor expansion as follows:

$$P(y|x, n^{(l,k,i)} = u) \approx P(y|x, n^{(l,k,i)} = \bar{u}) + \frac{\partial P(y|x, n^{(l,k,i)} = \bar{u})}{\partial \bar{u}} \times (u - \bar{u}) \tag{3}$$

When we set $u = 0$, we obtain the following formula:

$$s^{(l,k,i)}(x,y) = \bar{u} \times \frac{\partial P(y|x, n^{(l,k,i)} = \bar{u})}{\partial \bar{u}} \approx P(y|x, n^{(l,k,i)} = \bar{u}) - P(y|x, n^{(l,k,i)} = 0) \tag{4}$$

To calculate the causal effects of all neurons, we have to run the inference by masking each neuron one-at-a-time, which requires an enormous computational cost because LLMs typically contain millions of neurons (Table 12). In contrast, we can efficiently calculate $s^{(l,k,i)}(x,y)$ in a single run.

We aggregate the score on a dataset $D$ with $Q$ instances as the weighted sum over the exact probability:

$$s^{(l,k)}(D) = \sum_{q=1}^{Q} P(y_q|x_q) \times s^{(l,k)}(x_q, y_q) \tag{5}$$

This is because when the model predicts the correct answer with higher confidence, it should contain more reliable information.

## 3.3 Neuron Selection

To identify *culture-general* and *culture-specific* neurons, we use the MCQs from BLEnD, which provide sufficient instances and reduce the risk of overfitting to individual examples. BLEnD covers 16 countries and six categories, ensuring diversity in cultural topics. To test whether identified neurons generalize across different domains of cultural knowledge, we split BLEnD by category: three categories (*food*, *work-life*, *sport*) for neuron identification (BLEnD$_{\text{neur}}$) and the remaining three (*education*, *family*, *holidays/celebrations/leisure*) for evaluation (BLEnD$_{\text{test}}$). BLEnD provides 500 questions, and each question has multiple instances derived from different answer choices. We sample up to five instances per question to balance the number of instances of each question, yielding 12,701 instances in BLEnD$_{\text{neur}}$ and 10,331 in BLEnD$_{\text{test}}$.

Table 1: An example of the CountryRC (CRC) dataset.

| Passage | Question |
|---|---|
| Matthew applied for internships in both {country_A} and {country_B}, but only the company in {country_A} responded. He accepted and worked there over the summer. | Which country did Matthew go to for his internship? A. ... |

Moreover, we prepare $\text{BLEnD}_{\text{ctrl}}$ to isolate neurons that contribute purely to cultural inference. In $\text{BLEnD}_{\text{ctrl}}$, the question content is removed, leaving only the answer choices and the instruction for the answer format (Table 4). Neuron scores are calculated as $s^{(l,k)}(\text{BLEnD}_{\text{neur}}) - s^{(l,k)}(\text{BLEnD}_{\text{ctrl}})$, so that we can exclude neurons related to other properties, such as task understanding.

Since BLEnD evaluates culturally dependent knowledge, all the problems explicitly include country names. Thus, the top-scoring neurons on $\text{BLEnD}_{\text{neur}}$ may contain superficial neurons that simply respond to tokens of country names rather than cultural content. To filter out such superficial neurons, we construct another control dataset called CountryRC[1] (CRC), in which the correct answer is always a country name that appears in the context (Table 1). We utilize ChatGPT[2] to create CRC. Answering CRC requires models to recognize and propagate information about the country name, but it does not involve cultural understanding. CRC contains 50 problems per country, with half used for neuron identification ($\text{CRC}_{\text{neur}}$), and the remainder for evaluation ($\text{CRC}_{\text{test}}$).

For *culture-general* neurons, we first select the top $t\%$ of neurons ranked by $s^{(l,k)}(\text{BLEnD}_{\text{neur}}) - s^{(l,k)}(\text{BLEnD}_{\text{ctrl}})$, and then exclude the top $r\%$ ranked by $s^{(l,k)}(\text{CRC}_{\text{neur}})$. This procedure defines CULNIG-general. For *culture-specific* neurons of a country $c$, we first apply the same process to select neurons using only the instances of $c$ in the datasets, with an additional filtering step. Specifically, the score for $c$ is calculated as $s^{(l,k,c)} = s^{(l,k)}(\text{BLEnD}_{\text{neur}}^{(c)}) - s^{(l,k)}(\text{BLEnD}_{\text{ctrl}}^{(c)})$. We compute the z-score of each neuron over the 16 countries in BLEnD as $z^{(c)} = \frac{s^{(l,k,c)} - \mu}{\sigma}$, where $\mu$ and $\sigma$ are the mean and standard deviation of $s^{(l,k,c)}$ across countries. Neurons with $z^{(c)} < 0.5$ are removed, as they are likely to contribute to multiple cultures. This threshold of z-score is determined through a preliminary experiment. The whole pipeline defines CULNIG-specific.

## 4 EXPERIMENT AND ANALYSIS

In this section, we first describe our experimental settings in Section 4.1. We then compare the roles of each module and decide the thresholds in Section 4.2. Based on its result, we identify *culture-general* neurons in Section 4.3 and *culture-specific* neurons in Section 4.4. Next, we perform further analysis about what these neurons encode and compute in Section 4.5. Finally, we show a potential application of our findings from an engineering perspective in Section 4.6.

### 4.1 MODELS AND DATASETS

In our experiments, we use gemma-3-12b-it, gemma-3-27b-it (Gemma Team, 2025), Qwen-3-14B (Qwen Team, 2025), Llama-3-8B-Instruct (Grattafiori et al., 2024), phi-4 (Abdin et al., 2024), and Falcon3-10B-Instruct (TII Team, 2024). We select various state-of-the-art open-source models to demonstrate the robustness and generalizability of our findings (see Appendix B for details).

As explained in Section 3.3, we use $\text{BLEnD}_{\text{neur}}$, $\text{BLEnD}_{\text{ctrl}}$, and $\text{CRC}_{\text{neur}}$ for neuron identification. For evaluation, we use $\text{BLEnD}_{\text{test}}$ and CulturalBench (CultB) to measure cultural knowledge, NormAd as a task involving both cultural knowledge and values, and WorldValuesBench (WVB) to assess the understanding of cultural values. We also use short answer questions (SAQs) of $\text{BLEnD}_{\text{test}}$ to evaluate LLMs in a different task and in multilingual settings. In addition, we utilize four NLU benchmarks: $\text{CRC}_{\text{test}}$, CommonsenseQA (ComQA) (Talmor et al., 2019), QNLI, and MRPC (Wang et al., 2019), as comparison tasks that do not necessarily require cultural understanding.

---

[1] https://huggingface.co/datasets/Taise228/CountryRC
[2] https://chatgpt.com/overview

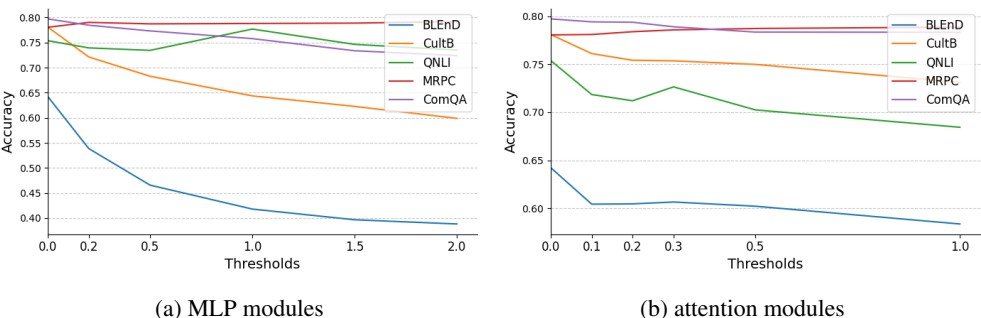

(a) MLP modules          (b) attention modules

Figure 2: Accuracy of gemma-3-12b-it on each benchmark as more top-scoring neurons on BLEnD (threshold $t$) are masked, with neurons selected from MLP and attention modules, respectively.

Regarding the evaluation metrics, we use accuracy (%) for all benchmarks except WVB. For WVB, we frame the task as a prediction of a questionnaire response given the country. The questionnaire uses a Likert scale, and we adopt the $\text{score}_c$ metric following Xu et al. (2025):

$$\text{score}_c = \frac{1}{N} \sum_{n=1}^{N} \Big(1 - \frac{|a_c^{(n)} - p_c^{(n)}|}{\text{max distance}}\Big) \times 100. \tag{6}$$

Here, $a_c^{(n)}$ is the majority answer among participants from country $c$, $p_c^{(n)}$ is the model prediction, and max distance is the maximum possible distance between the options and $a_c^{(n)}$. A higher $\text{score}_c$ indicates a stronger alignment.

Taking into account the sensitivity of LLMs to task instructions (Zhan et al., 2024), we prepare four prompt formats for each benchmark using ChatGPT (for BLEnD, the task instruction is included in the questions, so we prompt them without additional instructions). More details are given in Appendix A.

## 4.2 ROLES OF MODULES: ATTENTION VS MLP

First, we conduct a preliminary experiment to analyze the roles of each module and decide the threshold in CULNIG-general. We separately select neurons from MLP and attention modules of gemma-3-12b-it, varying the threshold $t$ for the top-ranked neurons. We fix the threshold for $\text{CRC}_{\text{neur}}$ to $r = 1\%$. Figure 2 shows the evaluation results when masking the identified neurons.

We find that masking MLP neurons causes substantial degradation on cultural benchmarks, while accuracies on QNLI and MRPC remain unaffected. For ComQA, the accuracy shows a moderate drop, likely because ComQA contains culture-related questions (e.g., `What island country is ferret popular?` → `great britain`). Beyond $t = 1\%$, declines on BLEnD$_{\text{test}}$ and CultB become gradual and parallel those on QNLI and ComQA, indicating that additional neurons contribute less specifically to cultural understanding. For attention neurons, the overall impact is smaller, but the scores on cultural benchmarks and QNLI decline to some extent. Although QNLI is solvable only with in-context information, it contains cultural sentences (e.g., `What is the first major city in the stream of the Rhine?`). Cultural knowledge can help solve QNLI, so the reduction may come from lost cultural understanding. Therefore, attention modules can still contain culture neurons. Beyond $t = 0.2\%$, the slopes on cultural benchmarks are similar to those on QNLI and ComQA.

These results corroborate prior studies showing that transformer MLPs primarily support knowledge recall, whereas attention modules facilitate in-context information processing (Meng et al., 2022; Ortu et al., 2024). This observation suggests that LLMs can rely more heavily on MLP neurons to solve cultural benchmarks, which require recall of out-of-context knowledge. Based on these observations, we adopt different thresholds for MLP and attention neurons in CULNIG-general, setting $t_{\text{MLP}} = 1\%$ and $t_{\text{attn}} = 0.2\%$. These thresholds are further validated by the sensitivity analysis described in Appendix C. In CULNIG-specific, we do not separate MLP and attention neurons, since the z-score-based filtering step can remove neurons that facilitate task understanding.

Table 2: Evaluation results of masking *culture-general* (cult) and random (rand) neurons. Random scores are averaged over ten seeds of neuron selection. Values in parentheses denote standard deviations. **Bold** values indicate statistically significant score reductions relative to the random scores.

| Model | | #Neuron | BLEnD$_{test}$ | CultB | NormAd | WVB | ComQA | QNLI | MRPC |
|---|---|---|---|---|---|---|---|---|---|
| Chance rate | | - | 25.00 | 25.00 | 33.33 | 49.85 | 20.00 | 50.00 | 50.00 |
| gemma-3 -12b-it | orig | 0 | 64.22 | 78.08 | 58.54 | 64.08 | 79.71 | 75.37 | 78.04 |
| | cult | 8,087 | **37.93** | **62.00** | **52.02** | **58.46** | **75.10** | 72.77 | 78.65 |
| | rand | 8,087 | 63.57(0.46) | 77.31(0.28) | 57.55(0.57) | 64.03(0.59) | 79.18(0.60) | 75.46(4.81) | 78.22(0.53) |
| gemma-3 -27b-it | orig | 0 | 61.37 | 81.32 | 58.76 | 64.47 | 80.88 | 91.43 | 78.30 |
| | cult | 14,273 | **39.96** | 69.76 | **52.31** | 60.98 | 79.32 | 90.81 | 78.86 |
| | rand | 14,273 | 62.17(3.15) | 78.32(7.11) | 57.19(1.62) | 62.07(7.03) | 78.40(6.71) | 87.56(9.87) | 77.21(2.64) |
| Qwen3 -14B | orig | 0 | 65.96 | 76.92 | 56.85 | 65.22 | 81.76 | 71.31 | 79.91 |
| | cult | 7,340 | **35.84** | **57.07** | **49.02** | **60.70** | **75.23** | 76.20 | **78.70** |
| | rand | 7,340 | 65.47(0.49) | 75.98(0.40) | 56.26(0.65) | 64.46(1.04) | 80.86(0.42) | 71.49(1.2) | 79.64(0.42) |
| Llama- 3.1-8B- Instruct | orig | 0 | 60.18 | 70.54 | 47.71 | 64.05 | 76.74 | 64.43 | 73.93 |
| | cult | 4,268 | **32.19** | **36.94** | **37.65** | **51.68** | **51.97** | 48.64 | 69.35 |
| | rand | 4,268 | 57.75(0.97) | 67.25(1.03) | 43.88(1.59) | 61.55(1.71) | 72.84(1.24) | 55.78(6.05) | 70.49(2.26) |
| phi-4 | orig | 0 | 63.89 | 78.30 | 59.68 | 65.0 | 80.43 | 89.15 | 78.57 |
| | cult | 7,447 | **35.05** | **57.72** | 51.84 | 66.48 | **70.60** | 85.84 | 77.00 |
| | rand | 7,447 | 63.29(0.63) | 76.94(1.71) | 56.38(2.98) | 61.82(2.67) | 78.89(2.10) | 86.98(1.93) | 76.04(2.47) |
| Falcon3 -10B- Instruct | orig | 0 | 57.98 | 71.74 | 55.26 | 58.00 | 79.73 | 74.57 | 78.59 |
| | cult | 9,282 | **35.47** | **56.81** | **48.75** | 59.16 | **71.85** | 70.30 | 78.43 |
| | rand | 9,282 | 57.64(0.31) | 71.07(0.23) | 54.06(1.39) | 57.4(0.83) | 78.89(0.71) | 74.17(3.19) | 78.56(0.19) |

We set $t = 0.3\%$ and $r = 1\%$, reflecting the expectation that *culture-specific* neurons are fewer than *culture-general* neurons.

## 4.3 CULTURE-GENERAL NEURONS

With the settings described in Section 4.2, we identify *culture-general* neurons. Table 2 shows the evaluation results when suppressing *culture-general* neurons and random neurons averaged over ten seeds. In the table, p-values are defined as the probability that the score reduction with random neurons is greater than or equal to that with *culture-general* neurons, estimated by setting a bootstrapping sample size to 2,000. Here, the scores of random neurons are computed in two different ways: (a) as the average over ten seeds and (b) as the score of a uniformly sampled single seed (for sensitivity analysis). If both p-values are smaller than 0.05, *culture-general* neurons are regarded as statistically significant.

We observe that eliminating *culture-general* neurons consistently causes significant degradation on cultural benchmarks, while the impact on NLU benchmarks is smaller. In particular, for BLEnD$_{test}$, the score drops substantially up to 30%, although the identified neurons account for fewer than 1% of the total. For CRC$_{test}$, the models achieved almost 100% accuracy both before and after masking neurons (Table 23), suggesting that few superficial neurons have been included. Notably, although neurons are identified solely using specific cultural knowledge categories, performance also declines on the unseen categories (BLEnD$_{test}$), demonstrating generalization beyond knowledge domains. This generalization further extends across task formats (CultB) and even across cultural attributes, such as cultural etiquette and values (NormAd and WVB). Moreover, we evaluate the models on SAQs of BLEnD$_{test}$ and demonstrate that masking *culture-general* neurons degrades the accuracy in the multilingual setting as well (Table 15, Appendix E). These results imply that *culture-general* neurons capture a broad representation of cultural understanding.

Figure 3 shows the distribution of *culture-general* neurons in gemma-3-12b-it. Most of the neurons are located in shallow to middle MLP modules. This trend is consistent across models (Appendix D), suggesting that CULNIG-general captures a general property of LLMs. We also show the ablation studies of each step in CULNIG-general in Appendix K.

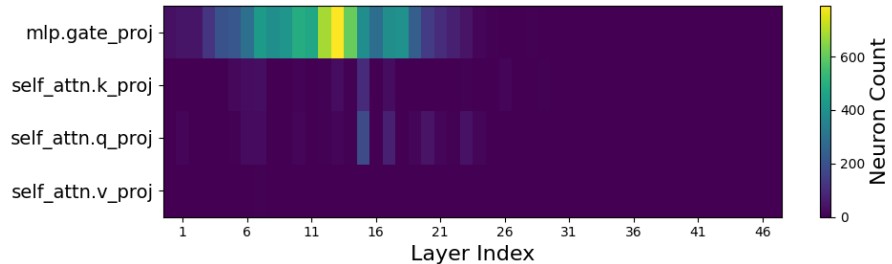

Figure 3: The distribution of *culture-general* neurons in gemma-3-12b-it.

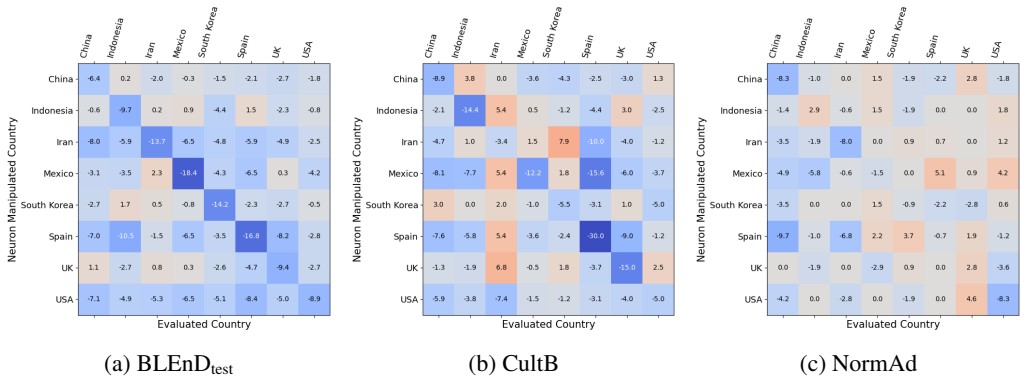

| (a) BLEnD_test | (b) CultB | (c) NormAd |

Figure 4: Score reductions after masking *culture-specific* neurons of gemma-3-12b-it.

## 4.4 CULTURE-SPECIFIC NEURONS

Next, we apply CULNIG-specific to identify *culture-specific* neurons that support understanding of individual cultures. We focus on eight countries covered in all of BLEnD, CultB, and NormAd (China, Indonesia, Iran, Mexico, South Korea, Spain, UK, and USA), which are culturally diverse in the Inglehart-Welzel World Cultural Map from WVS Wave 7 (Haerpfer et al., 2020).

Figure 4 shows score reductions when masking *culture-specific* neurons in gemma-3-12b-it. For BLEnD_test and CultB, the largest drops occur in the target cultures, confirming that identified neurons are associated with knowledge of the target culture. Moreover, *culture-specific* neurons tend to affect related cultures. For example, masking Mexico-specific neurons most strongly affects the problem instances of Mexico (the mean rank of score reduction among 16 cultures over six models is 1.17), and the second most affected culture is Spain (the mean rank was 3.83). We observe that historically or geographically related cultures tend to affect each other, indicating that the neurons underlying the related cultures are shared (Table 17, Table 18, Table 19). In contrast, these patterns are less clear for NormAd, which suggests that *culture-specific* neurons capture less etiquette and values.

The distribution of *culture-specific* neurons is similar to that of *culture-general* neurons (Figure 12a). In contrast, these results differ from CAPE, which reported that culture neurons are concentrated in the upper layers. We replicated the experiments of CAPE with gemma-3-12b-it, but failed to reproduce it. Consequently, LAPE and CAPE neurons had negligible impacts on evaluation scores. Further investigation of this discrepancy is left for future work. Possible factors for the difference of the distributions are differences in attribution scores (gradient-based in ours and activation-based in CAPE) and evaluation metrics (QA accuracy in ours and perplexity in CAPE). Additionally, recent studies have demonstrated that LLMs process multilingual prompts through three stages: (1) map multilingual inputs into the shared representation at the early layers, (2) process semantic information in the shared space at the middle layers, and (3) translate back for generation at the upper layers (Wang et al., 2025; Wu et al., 2025). Thus, we can hypothesize that CAPE neurons, which are located in the very early and upper layers, primarily respond to token patterns specific to the target culture. A detailed information on this evaluation is presented in Appendix G.

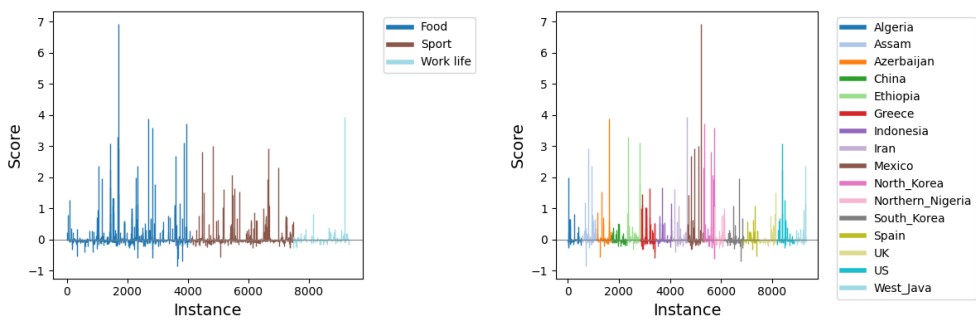

(a) Colored by question categories.

(b) Colored by question target cultures.

Figure 5: Attribution scores of the top scoring *culture-general* neurons in gemma-3-12b-it ($1141^{st}$ neuron in attention query projection in the 15th layer) per instance in $\text{BLEnD}_{\text{neur}}$.

### 4.5 INSTANCE-LEVEL ANALYSIS OF NEURON SCORES

Now that we have identified *culture-general* and *culture-specific* neurons, the following question arises: *Do these neurons encode meta-level control signals for cultural contexts or knowledge-level concepts?* To give insight into this point, we examine the instance-level neuron attribution scores. We sample several *culture-general* and *culture-specific* neurons and plot their scores (Equation 2) per instance in $\text{BLEnD}_{\text{neur}}$ where the model assigns the correct probability$>0.5$. If a neuron encodes meta-level control signals, it should have high scores on most instances.

Figure 5 shows the instance-level scores of the top-scoring *culture-general* neuron in gemma-3-12b-it. We observe that the scores take positive values on only $29\%$ of instances and that high-scoring instances exist across categories and target cultures. It indicates that the neuron encodes knowledge-level concepts rather than meta-level signals and that various kinds of concepts are encoded regardless of their types. This trend is common in other neurons (Appendix H). Together with Table 2, it is suggested that *culture-general* neurons mainly encode cultural concepts, including knowledge and values, but not general commonsense knowledge or NLU. On the other hand, the scores of *culture-specific* neurons tend to be especially high on their own and related cultures (Figure 30). For example, the top Iran-specific neuron has high scores on instances of Iran and Azerbaijan. Meanwhile, only $38\%$ of Iranian instances have positive scores for that neuron. Therefore, *culture-specific* neurons may predominantly encode knowledge of specific cultures, but not at the meta-level.

To further investigate the roles of neurons, we split NormAd into $\text{NormAd}_{\text{neur}}$ and $\text{NormAd}_{\text{test}}$ and use $\text{NormAd}_{\text{neur}}$ to calculate the attribution scores in CULNIG-general instead of $\text{BLEnD}_{\text{neur}}$. As a result, masking neurons identified with $\text{NormAd}_{\text{neur}}$ degrades the scores on all cultural benchmarks. Additionally, when we exclude neurons identified with $\text{NormAd}_{\text{neur}}$ from those with $\text{BLEnD}_{\text{neur}}$, masking the neurons still has a negative impact on $\text{NormAd}_{\text{test}}$ and WVB. These results suggest that a neuron with high scores on $\text{BLEnD}_{\text{neur}}$ can contribute to $\text{NormAd}_{\text{test}}$ even if it does not have high scores on $\text{NormAd}_{\text{neur}}$, validating that cultural knowledge and values can be encoded in the same neurons. Further details are provided in Appendix I.

### 4.6 APPLICATIONS: TARGET MODULE SELECTION FOR TRAINING

In this section, we demonstrate a potential application of our findings from an engineering perspective. Fine-tuning LLMs often risks degrading their abilities on other tasks (Luo et al., 2025) and also requires enormous computational costs. To achieve robust and efficient training, we propose to select updating modules based on their roles.

We fine-tune gemma-3-12b-it with QNLI and MRPC, updating only a portion of the modules. For module selection, we sort the modules by the number of *culture-general* neurons, and select either those with the most *culture-general* neurons (top-culture modules) or those with none (bottom-culture modules) until the number of parameters exceeds 10%. When an MLP gate projection is selected, we also include the corresponding up and down projections, and when a query, key, or value module is selected, we also include the corresponding query, key, value, and out projections,

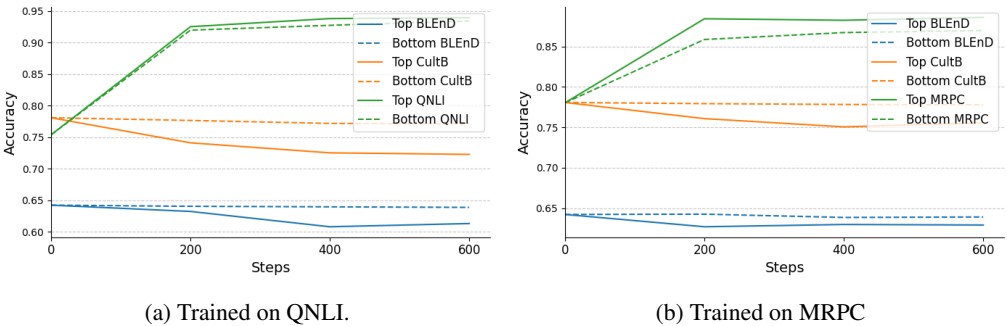

(a) Trained on QNLI.
(b) Trained on MRPC

Figure 6: Evaluation results of gemma-3-12b-it on BLEnD$_{test}$, CultB, QNLI, and MRPC when fine-tuned on QNLI or MRPC, updating only 10% of the total parameters. Updated modules are selected either from those containing the most *culture-general* neurons (Top) or those without *culture-general* neurons (Bottom).

since neurons in those modules are connected as subkeys and subvalues (Section 3.1). We fine-tune the model for 600 steps with a learning rate of 3e-5 and evaluate it every 200 steps.

The selected top-culture modules are all MLP modules from shallow to middle layers, while the bottom-culture modules mainly consist of very shallow attention modules and very deep attention and MLP modules (Table 22). The evaluation results are shown in Figure 6. We observe that target scores (QNLI or MRPC) improve in both cases. However, when updating the top-culture modules, the scores of cultural benchmarks decrease. Meanwhile, updating the bottom-culture modules has little effect on cultural abilities. These results suggest that we can train the model efficiently and robustly by selecting target components based on their roles. Further experiments with different parameter settings are shown in Appendix J.

Although our experiments are limited to QNLI and MRPC, these benchmarks are well-established and widely used for measuring NLU of LLMs. Our results could also be applied to other NLP tasks. Moreover, we believe that our findings can be used to improve the cultural understanding of LLMs. For example, when we use knowledge editing methods (e.g., Meng et al. (2023), Fang et al. (2025)) to update cultural knowledge, we can target the top-culture modules. In addition, when we train a model to insert new cultural knowledge, we can update the neurons that are not included in *culture-general* neurons in the top-culture modules so that new knowledge is easily incorporated while retaining existing knowledge. Actual applications are deferred to future work.

## 5 CONCLUSION

We introduced CULNIG to identify neurons that contribute to the cultural understanding of LLMs. We evaluated six LLMs with *culture-general* neurons masked and demonstrated that the scores on the cultural benchmark decreased significantly, while the impacts on the NLU benchmarks were minor. Despite being identified from a limited set of cultural knowledge problems, these neurons affected broader cultural attributes, including the understanding of cultural knowledge in multilingual settings and cultural values. In addition, we discovered *culture-specific* neurons that are tied to individual cultures and showed that masking those neurons degraded knowledge of the target and related cultures. *Culture-general* and *culture-specific* neurons are concentrated in shallow to middle MLP layers. Finally, we demonstrated that when we fine-tuned LLMs on NLU benchmarks, cultural understanding was more easily lost by updating modules containing many *culture-general* neurons than by updating modules without *culture-general* neurons. We believe that our findings provide a foundation for future studies to improve the cultural understanding of LLMs.

REPRODUCIBILITY STATEMENT

For reproducibility, we release our code at `https://github.com/ynklab/CULNIG`. In the repository, we include the scripts of CULNIG to identify *culture-general* and *culture-specific* neurons, the script for evaluation, the training script for the experiment in Section 4.6, the prompts, and the CRC dataset. For detailed information and the usage of the scripts, refer to `README.md` in the repository.

ACKNOWLEDGMENTS

We thank the four anonymous reviewers and the meta-reviewer for their helpful comments and feedback. We also thank Daiki Matsuoka, Koki Ryu, Xiaotian Wang, Rongzhi Li, and Tomoki Doi for their insightful discussions. This work was supported by JST CREST Grant Number JPMJCR2565 and JST BOOST Grant Number JPMJBY24H5, Japan. Danushka Bollegala holds concurrent appointments as a Professor at the University of Liverpool and as an Amazon Scholar at Amazon. This paper describes work performed at the University of Liverpool and is not associated with Amazon.

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

# A DATASET DETAILS

Table 3: Cultural benchmarks used in our experiments.

| Benchmark | #Country | #Instance | Target |
|---|---|---|---|
| BLEnD[3] (Myung et al., 2024) | 16 | 306k MCQs, 15k SAQs | Everyday knowledge in a diverse culture |
| CulturalBench[4] (Chiu et al., 2025) | 45 | 1.23k | Cultural knowledge |
| NormAd[5] (Rao et al., 2025) | 75 | 2.63k | Cultural etiquette and norms |
| WorldValuesBencb[6] (Zhao et al., 2024) | ~64 | 260 per participants | Cultural values |

Table 3 lists the cultural benchmarks used in our experiments. As explained in Section 2.1, BLEnD evaluates everyday cultural knowledge of LLMs in multiple-choice questions (MCQs) and short answer questions (SAQs). CulturalBench has two task formats: CulturalBench-Easy, which asks about cultural knowledge in multiple-choice questions with four options, and CulturalBench-Hard, which asks whether each of these four options is correct or not with the same question. For simplicity, we adopt CulturaBench-Easy for evaluation. In NormAd, a model is asked to determine whether a given daily scenario is acceptable in a specified culture, which requires understanding of both cultural knowledge and values. The task of WorldValuesBench is to predict participants' responses to questionnaires given their demographic information. Questionnaires are common for all participants, such as `do you believe in God?`, derived from World Values Survey Wave 7. We download the datasets from Hugging Face Datasets and the GitHub repositories.

Table 4: Examples of $BLEnD_{neur}$ and $BLEnD_{ctrl}$.

| $BLEnD_{neur}$ | $BLEnD_{ctrl}$ |
|---|---|
| What is a common snack for preschool kids in the UK? Without any explanation, choose only one from the given alphabet choices(e.g., A, B, C). Provide as JSON format: {"answer_choice":""} A. cookie B. egg C. fruit D. jelly Answer: | Without any explanation, choose only one from the given alphabet choices(e.g., A, B, C). Provide as JSON format: {"answer_choice":""} A. cookie B. egg C. fruit D. jelly Answer: |

As described in Section 3.3, we prepare $BLEnD_{ctrl}$ corresponding to each question of $BLEnD_{neur}$. Table 4 shows examples of $BLEnD_{neur}$ and $BLEnD_{ctrl}$. $BLEnD_{ctrl}$ is created by omitting the question content from the instances of $BLEnD_{neur}$. In CULNIG, by subtracting the neuron attribution score of $BLEnD_{ctrl}$ from that of $BLEnD_{neur}$, we can measure the sheer contribution of neurons to culture knowledge.

Moreover, we constructed the CountryRC (CRC) dataset to filter out superficial neurons that respond to country names. We utilized ChatGPT to create CRC. We instructed ChatGPT to generate reading comprehension problems that contain a country name in their context, and the answer is that country name. We also specified that the problems must not require any cultural understanding. CRC has 50 instances, and 30 instances have only one country name in their context, and the remaining 20 contain an additional dummy country name. Each instance has four answer choices of country names. Country names are represented as placeholders and replaced with the actual names of the target countries. CRC has been released in our repository.

---

[3] https://huggingface.co/datasets/nayeon212/BLEnD
[4] https://huggingface.co/datasets/kellycyy/CulturalBench
[5] https://huggingface.co/datasets/akhilayerukola/NormAd
[6] https://github.com/Demon702/WorldValuesBench/tree/635db7455e2c656978929210eba984bc09ddd659

In addition to the cultural benchmarks, we use NLU benchmarks that do not necessarily require cultural understanding for evaluation. If masking identified neurons results in the reduction of NLU abilities, we cannot conclude that the identified neurons really support the cultural mechanisms of an LLM, even if masking them degrades cultural understanding. We use CommonsenseQA[7], QNLI[8], and MRPC[9]. As for CommonsenseQA and QNLI, ground truths for the test set have not been published, so we use the validation set for evaluation.

Considering the sensitivity of LLMs to prompt wording, we prepared four task instructions for each evaluation dataset except for BLEnD. For BLEnD, task instructions are already included in data sources, and each problem has multiple instances with diverse answer choices, so we used them without additional instructions. We used the prompts in the original paper as a seed and utilized ChatGPT to rephrase the prompts. We show the prompts for each dataset in Table 5, Table 6, Table 7, Table 8, Table 9, Table 10, and Table 11.

Table 5: Evaluation prompts for CulturalBench

| Instruction |
| --- |
| To answer the following multiple-choice question, you should choose one option only among A,B,C,D. Do not output any other things.
Question: {question}
A. {option_a}
B. {option_b}
C. {option_c}
D. {option_d} |
| Select only one option from A, B, C, or D to answer the following multiple-choice question. Do not output anything else.
Question: {question}
A. {option_a}
B. {option_b}
C. {option_c}
D. {option_d} |
| Choose one answer among A, B, C, and D for the question below. Do not include any explanation or extra content.
Question: {question}
A. {option_a}
B. {option_b}
C. {option_c}
D. {option_d} |
| You must answer the following question by selecting a single choice from A, B, C, or D. Do not write anything else.
Question: {question}
A. {option_a}
B. {option_b}
C. {option_c}
D. {option_d} |

[7]https://huggingface.co/datasets/tau/commonsense_qa
[8]https://huggingface.co/datasets/nyu-mll/glue
[9]https://huggingface.co/datasets/nyu-mll/glue

Table 6: Evaluation prompts for NormAd

| Instruction |
| --- |
| You will be given a short story involving 2 or more characters about everyday situations. You will also be given a country of relevance. Read the story and determine if the character's behavior aligns with the social norms of that country. Only consider the provided context and country. Do not include any outside assumptions. Respond **only** with the number 1, 2, or 3.

country: {country}
Story: {story}
Options: {options} |
| Read a short story describing everyday events with at least two characters. You'll also receive a country. Your task is to judge whether the character's actions are in line with the cultural expectations of that country. Use only the given story and cultural context; do not bring in any outside knowledge. Answer strictly with the number 1, 2, or 3.

country: {country}
Story: {story}
Options: {options} |
| A short scenario featuring multiple characters will be provided, along with the cultural background of a specific country. Your role is to determine how well the behavior shown fits that country's social norms. Do not make inferences beyond the given content. Only respond with 1, 2, or 3.

country: {country}
Story: {story}
Options: {options} |
| Given a short daily-life story involving multiple characters, along with the country, assess whether the behavior depicted fits within the social norms of that culture. Stick strictly to the given material without adding outside reasoning. Answer with just 1, 2, or 3.

country: {country}
Story: {story}
Options: {options} |

Table 7: Evaluation prompts for WorldValuesBench

| Instruction |
| --- |
| System: You are a person from {country}.
Prompt: Question: {question}
Please respond with a single digit only from {min_option} to {max_option}. Do not include any other text. |
| System: You are a person from {country}.
Prompt: Question: {question}
Your answer should be a single digit between {min_option} and {max_option}. Do not add any other information. |
| System: Behave as if you are from {country}.
Prompt: Question: {question}
Please respond with a single digit only from {min_option} to {max_option}. Do not include any other text. |
| System: Behave as if you are from {country}.
Prompt: Question: {question}
Your answer should be a single digit between {min_option} and {max_option}. Do not add any other information. |

Table 8: Evaluation prompts for CountryRC

| Instruction |
|---|
| Read the passage carefully and choose a single option from A, B, C, D to answer the question. Do not output any other text.

passage: {passage}
question: {question}
A. {option_a}
B. {option_b}
C. {option_c}
D. {option_d} |
| Read the following passage and question. Then, pick the most suitable answer from the four options. Only return the letter of your choice (A, B, C, or D).

passage: {passage}
question: {question}
A. {option_a}
B. {option_b}
C. {option_c}
D. {option_d} |
| From the information provided in the passage, choose the best answer to the question. You must select a single choice: 1, 2, 3, or 4, and do not include any other text.

passage: {passage}
question: {question}
1. {option_a}
2. {option_b}
3. {option_c}
4. {option_d} |
| Determine the correct answer to the question based on the content of the passage. Respond with one of the following: 1, 2, 3, or 4. No additional text is needed.

passage: {passage}
question: {question}
1. {option_a}
2. {option_b}
3. {option_c}
4. {option_d} |

Table 9: Evaluation prompts for CommonsenseQA

| Instruction |
|---|
| To answer the following multiple-choice question, you should choose one option only among A,B,C,D,E. Do not output any other things.
Question: {question}
A. {option_a}
B. {option_b}
C. {option_c}
D. {option_d}
E. {option_e} |
| Choose one answer among A, B, C, D, and E for the question below. Do not include any explanation or extra content.
Question: {question}
A. {option_a}
B. {option_b}
C. {option_c}
D. {option_d}
E. {option_e} |
| Pick one option only — A, B, C, D, or E — as the answer to the question below. Do not provide any additional text.
Question: {question}
A. {option_a}
B. {option_b}
C. {option_c}
D. {option_d}
E. {option_e} |
| Please choose one and only one of the following options (A, B, C, D, or E) to answer the question. Do not add anything else.
Question: {question}
A. {option_a}
B. {option_b}
C. {option_c}
D. {option_d}
E. {option_e} |

Table 10: Evaluation prompts for QNLI

| Instruction |
| --- |
| Determine whether the following context sentence contains enough information to answer the question.
Question: {question}
Context: {sentence}
Respond with:
0 if it does (entailment)
1 if it does not (not_entailment)
Only answer with 0 or 1. |
| Classify the relationship between the following question and context.
Question: {question}
Context: {sentence}
Label as:
0: entailment – the question is supported by the context
1: not_entailment – the question is not supported by the context
Please respond with either 0 or 1 only. |
| Read the question and the context.
Question: {question}
Context: {sentence}
If the context provides enough evidence to answer the question, return 0 (entailment).
If the context is insufficient or irrelevant, return 1 (not_entailment).
Your answer should be either 0 or 1. |
| Your task is to judge if the answer to the question can be found in the context.
Question: {question}
Context: {sentence}
Answer 0 for entailment, and 1 for not_entailment. Do not include any other text. |

Table 11: Evaluation prompts for MRPC

| Instruction |
| --- |
| Determine whether the following two sentences are paraphrases of each other in meaning.
Sentence 1: {sentence1}
Sentence 2: {sentence2}
Respond with:
1 – if they are paraphrases
0 – if they are not paraphrases
Only answer with 0 or 1. |
| You are given two sentences. Judge whether they express the same meaning, even if the wording is different.
Sentence 1: {sentence1}
Sentence 2: {sentence2}
Answer with 1 if they are paraphrases, and 0 if they are not.
Please respond using only 0 or 1. |
| A paraphrase means that two sentences convey the same information using different words or structure.
Sentence 1: {sentence1}
Sentence 2: {sentence2}
Decide whether these sentences are paraphrases.
Return 1 for paraphrase, 0 for not paraphrase.
Your answer must be either 0 or 1. |
| Compare the following two sentences. If they convey the same meaning regardless of differences in wording, classify them as paraphrases.
Sentence 1: {sentence1}
Sentence 2: {sentence2}
Respond with:
1 – if they are semantically equivalent (paraphrase)
0 – if they are not semantically equivalent
Only use 0 or 1 as your answer. |

## B    MODEL DETAILS

In our experiment, we analyze six open source state-of-the-art LLMs: gemma-3-12b-it[10], gemma-3-27b-it[11], Qwen-3-14B[12], Llama-3-8B-Instruct[13], phi-4[14], and Falcon3-10B-Instruct[15]. We apply our methods to these models to show the robustness and generalizability of our findings. We download the parameters from Hugging Face Hub. The architectures of these models are similar, as described in Section 3.1.

Table 12: The total number of neurons in each module of each model.

| Models | Total Neuron Count | | | |
|---|---|---|---|---|
| | MLP gate | Attention query | Attention key | Attention value |
| gemma-3-12b-it | 737,280 | 196,608 | 98,304 | 98,304 |
| gemma-3-27b-it | 1,333,248 | 253,952 | 126,976 | 126,976 |
| Qwen3-14B | 696,320 | 204,800 | 40,960 | 40,690 |
| Llama-3.1-8B-Instruct | 458,752 | 131,072 | 32,768 | 32,768 |
| phi-4 | 716,800 | 204,800 | 51,200 | 51,200 |
| Falcon3-10B-Instruct | 921,600 | 122,880 | 40,960 | 40,960 |

The total number of neurons in each model module is shown in Table 12. We only include the modules from which we select culture neurons (see Section 3.1). The number of neurons in an MLP gate module is $intermediate\_size \times num\_layer$, the number of neurons in an attention query module is $head\_dim \times num\_head \times num\_layer$, and the number of neurons in an attention key and value module is both $head\_dim \times num\_kv\_head \times num\_layer$. When using grouped-query attention of the group size $g$, $num\_kv\_head = num\_head \div g$.

---

[10]https://huggingface.co/google/gemma-3-12b-it
[11]https://huggingface.co/google/gemma-3-27b-it
[12]https://huggingface.co/Qwen/Qwen3-14B
[13]https://huggingface.co/meta-llama/Llama-3.1-8B-Instruct
[14]https://huggingface.co/microsoft/phi-4
[15]https://huggingface.co/tiiuae/Falcon3-10B-Instruct

## C   SENSITIVITY ANALYSIS TO THRESHOLDS

As described in Section 4.2, we set the thresholds for MLP and attention neurons in CULNIG-general to $t_{\text{MLP}} = 1\%$ and $t_{\text{attn}} = 0.2\%$. Since our claim that the cultural understanding of LLMs is driven by a sparse set of neurons accounting for less than $1\%$ of the total depends on these thresholds, we further conduct sensitivity analysis regarding $t_{\text{MLP}}$ and $t_{\text{attn}}$. We increase one of $t_{\text{MLP}}$ and $t_{\text{attn}}$, fix the other, and evaluate gemma-3-12b-it with the identified neurons masked.

The results are shown in Table 13 and Table 14. When increasing $t_{\text{MLP}}$ to 2% or 3%, while some cultural scores decrease, the scores on ComQA and QNLI degrade equally or more significantly. Furthermore, increasing $t_{\text{attn}}$ to 0.4% or 0.6% has only a small effect on the scores. These results suggest that the neurons added when the thresholds are increased do not play a specific role in the cultural domain, which aligns with the results presented in Figure 2. This analysis strengthens our argument that *culture-general* neurons account for only a sparse set of neurons.

Table 13: The evaluation results of gemma-3-12b-it when increasing $t_{\text{MLP}}$ while fixing $t_{\text{attn}}$ to $0.2\%$.

| $t_{\text{MLP}}$ | #Neuron | BLEnD$_{\text{test}}$ | CultB | NormAd | WVB | ComQA | QNLI | MRPC |
|---|---|---|---|---|---|---|---|---|
| 0% (orig) | 0 | 64.22 | 78.08 | 55.42 | 64.08 | 79.71 | 75.37 | 78.04 |
| 1% (ours) | 8,087 | 37.93 | 62.00 | 50.73 | 58.46 | 75.10 | 72.77 | 78.65 |
| 2% | 15,426 | 36.41 | 57.50 | 51.06 | 60.28 | 72.03 | 68.68 | 78.88 |
| 3% | 22,770 | 35.29 | 55.54 | 49.36 | 58.88 | 69.16 | 65.23 | 78.58 |

Table 14: The evaluation results of gemma-3-12b-it when increasing $t_{\text{attn}}$ while fixing $t_{\text{MLP}}$ to $1\%$.

| $t_{\text{attn}}$ | #Neuron | BLEnD$_{\text{test}}$ | CultB | NormAd | WVB | ComQA | QNLI | MRPC |
|---|---|---|---|---|---|---|---|---|
| 0% (orig) | 0 | 64.22 | 78.08 | 55.42 | 64.08 | 79.71 | 75.37 | 78.04 |
| 0.2% (ours) | 8,087 | 37.93 | 62.00 | 50.73 | 58.46 | 75.10 | 72.77 | 78.65 |
| 0.4% | 8,868 | 37.84 | 61.80 | 51.46 | 58.71 | 75.20 | 73.40 | 79.03 |
| 0.6% | 9.652 | 37.50 | 61.04 | 51.60 | 58.56 | 74.88 | 73.42 | 79.19 |

# D    CULTURE NEURON DISTRIBUTION

We show the distributions of *culture-general* neurons in each model in Figure 3, Figure 7, Figure 8, Figure 9, Figure 10, and Figure 11. We can observe that the neuron distributions are similar for all the models, concentrated in shallow to middle MLP layers. This result suggests that our method captures mechanisms shared across LLMs.

In addition, Figure 12a shows the distribution of Chinese *culture-specific* neurons in gemma-3-12b-it, and Figure 12b shows the distribution of Chinese neurons identified by CAPE (pure). While CULNIG-specific Chinese neurons are mainly located in shallow to middle MLP layers, similarly to CULNIG-general, CAPE Chinese neurons are concentrated in deeper layers.

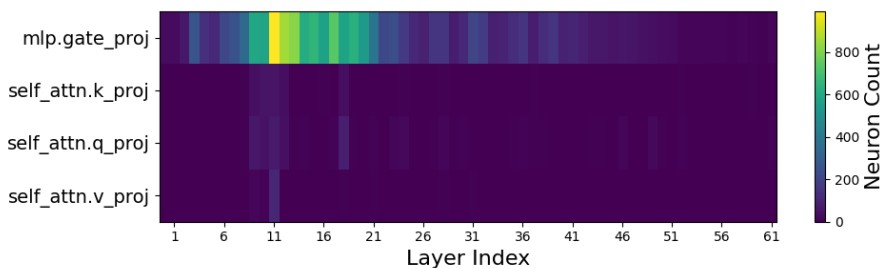

Figure 7: The distribution of *culture-general* neurons in gemma-3-27b-it.

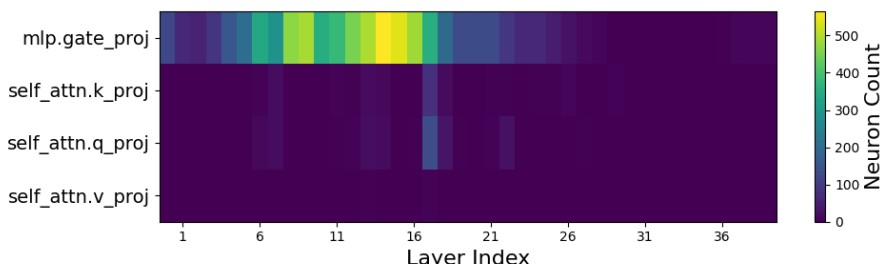

Figure 8: The distribution of *culture-general* neurons in Qwen3-14B.

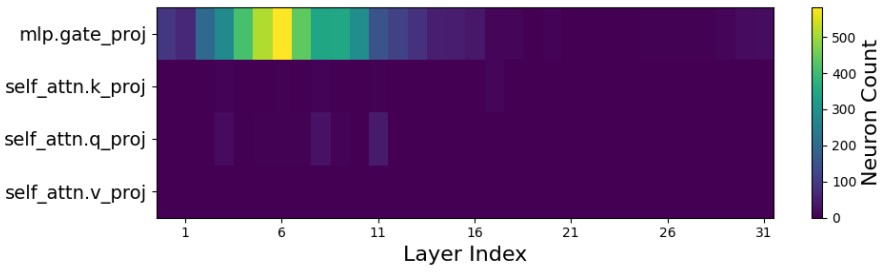

Figure 9: The distribution of *culture-general* neurons in Llama-3.1-8B-Instruct.

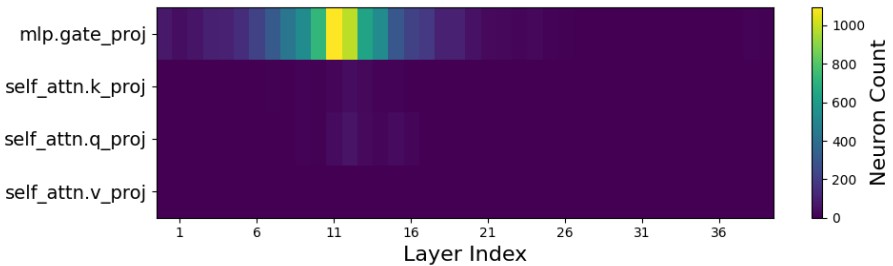

Figure 10: The distribution of *culture-general* neurons in phi-4.

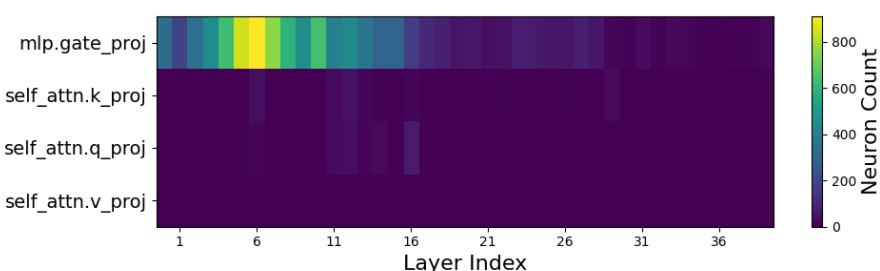

Figure 11: The distribution of *culture-general* neurons in Falcon3-10B-Instruct.

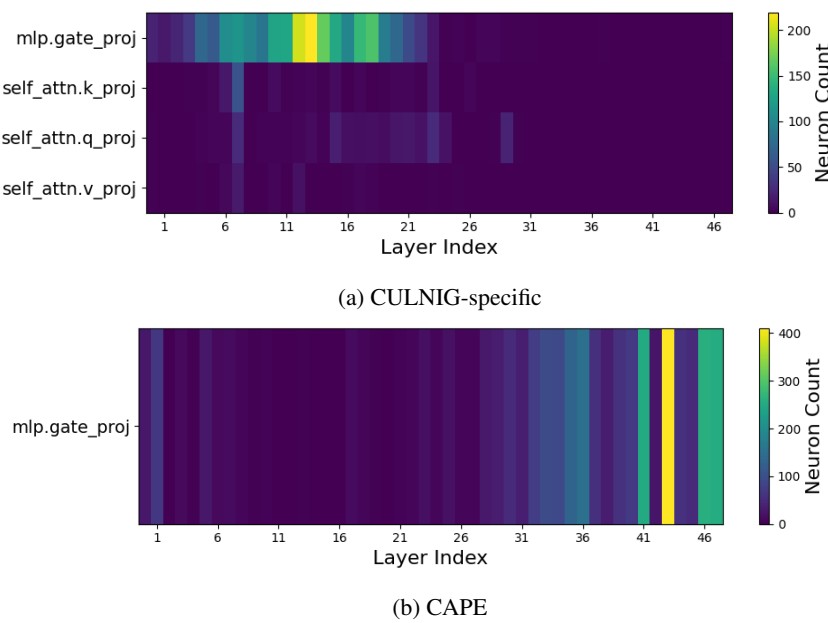

(a) CULNIG-specific

(b) CAPE

Figure 12: The distribution of Chinese neurons identified by CULNIG-specific and CAPE in gemma-3-12b-it.

# E RESULTS OF MULTILINGUAL EVALUATION ON BLEnD SAQ

As explained in Section 2.1, BLEnD provides two types of tasks: multilingual short answer questions (SAQs) and English multiple-choice questions (MCQs). The evaluation results on MCQs are shown in Section 4.3, confirming that suppressing *culture-general* neurons substantially degrades the performance of the models on MCQs (BLEnD$_{test}$). Here, we evaluate LLMs on BLEnD SAQs to see whether *culture-general* neurons are responsible for cultural understanding in multilingual and SAQ settings.

BLEnD covers 16 cultures, and the SAQs for each culture are provided in English and their corresponding language, resulting in 13 languages in total. We prompt LLMs only in their native language to evaluate each culture. Also, to align with the evaluation on MCQs, we use the same three categories as BLEnD$_{test}$. As for the task instruction, we utilize the prompts provided in their GitHub repository[16] and randomly select one instruction per instance. For other details of the evaluation, we follow the original settings of BLEnD (Myung et al., 2024) and their GitHub repositories. We set `max_new_tokens` to 512 and other parameters to the models' default values. When judging models' responses, we first lemmatize, stem, or tokenize the models' responses and the annotation answers. We regard the prediction as correct if any answers are included in the response.

Table 15: Evaluation accuracy (%) on BLEnD SAQs for the original model (Orig), when *culture-general* neurons are masked (Cult), and when random neurons are masked (Rand).

| Model | Orig | Cult | Rand |
|---|---|---|---|
| gemma-3-12b-it | 51.13 | 42.77 | 49.13 |
| gemma-3-27b-it | 57.71 | 47.00 | 56.32 |
| Qwen3-14B | 47.74 | 36.04 | 46.32 |
| Llama-3.1-8B-Instruct | 43.89 | 20.63 | 39.38 |
| phi-4 | 47.97 | 35.98 | 47.76 |
| Falcon3-10B-Instruct | 28.36 | 23.41 | 26.91 |

The accuracies on the SAQs are shown in Table 15. Suppressing *culture-general* neurons reduces the accuracy of all models more significantly than suppressing random neurons. Moreover, Figure 13, Figure 14, Figure 15, Figure 16, Figure 17, and Figure 18 show culture-wise accuracies of each model. We can observe that score reduction occurs regardless of cultures. These results indicate that *culture-general* neurons contribute to cultural understanding in multilingual and SAQ settings as well.

---

[16]https://github.com/nlee0212/BLEnD/tree/9972379c4fd20601691c45e6d7befa6a3eed7ed4

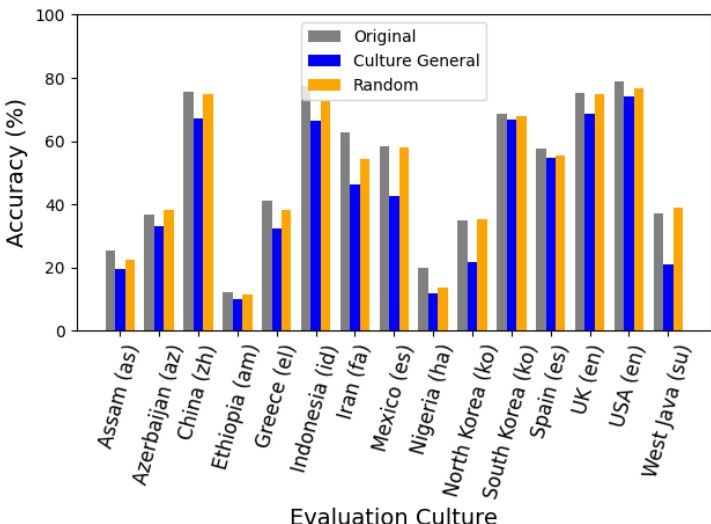

Figure 13: Accuracy of gemma-3-12b-it on BLEnD SAQs for each culture. Evaluation results of the original model, when masking *culture-general* neurons, and when masking random neurons are shown.

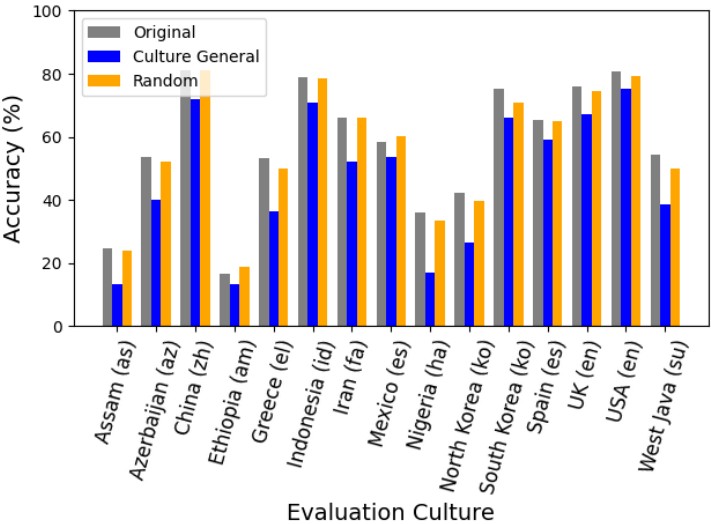

Figure 14: Accuracy of gemma-3-27b-it on BLEnD SAQs for each culture. Evaluation results of the original model, when masking *culture-general* neurons, and when masking random neurons are shown.

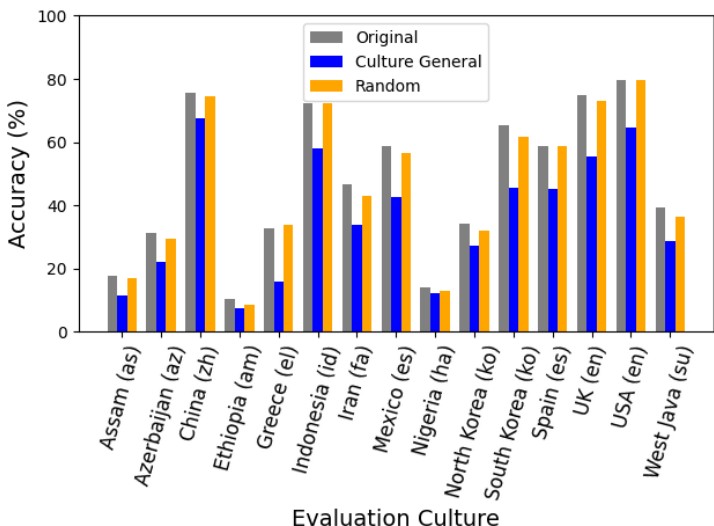

Figure 15: Accuracy of Qwen3-14B on BLEnD SAQs for each culture. Evaluation results of the original model, when masking *culture-general* neurons, and when masking random neurons are shown.

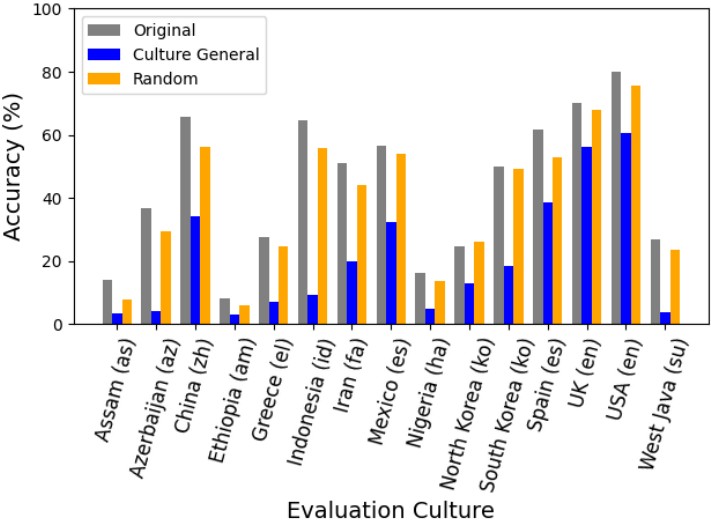

Figure 16: Accuracy of Llama-3.1-8B-Instruct on BLEnD SAQs for each culture. Evaluation results of the original model, when masking *culture-general* neurons, and when masking random neurons are shown.

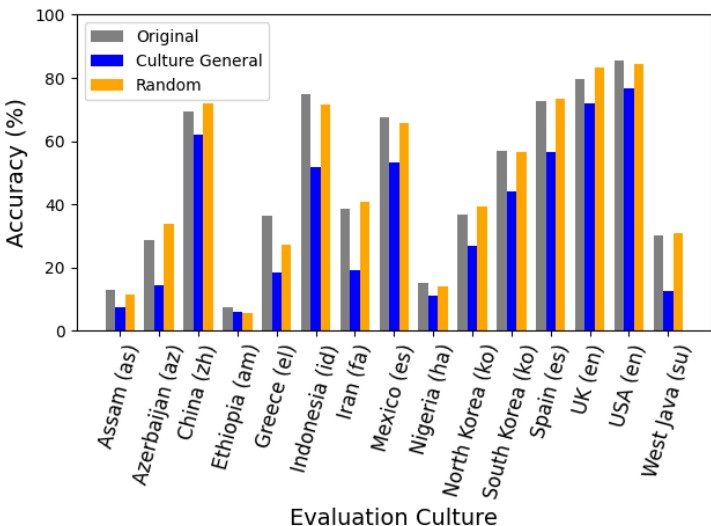

Figure 17: Accuracy of phi-4 on BLEnD SAQs for each culture. Evaluation results of the original model, when masking *culture-general* neurons, and when masking random neurons are shown.

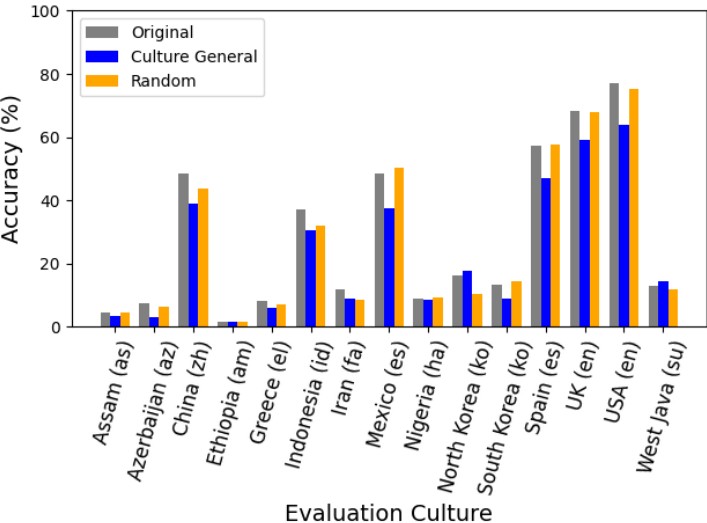

Figure 18: Accuracy of Falcon3-10B-Instruct on BLEnD SAQs for each culture. Evaluation results of the original model, when masking *culture-general* neurons, and when masking random neurons are shown.

## F    RESULTS OF CULTURE-SPECIFIC NEURONS

Table 16: The number of *culture-specific* neurons identified by CULNIG-specific.

| Model | China | Indo-nesia | Iran | Mex-ico | South Korea | Spain | UK | USA |
|---|---|---|---|---|---|---|---|---|
| gemma-3-12b-it | 2,667 | 2,569 | 2,948 | 2,756 | 2,101 | 2,655 | 2,977 | 3,041 |
| gemma-3-27b-it | 4,011 | 4,563 | 4,953 | 3,580 | 5,061 | 4,663 | 3,768 | 3,821 |
| Qwen3-14B | 1,553 | 1,897 | 1,473 | 2,070 | 2,204 | 1,874 | 2,029 | 2,190 |
| Llama-3.1-8B-Instruct | 471 | 782 | 549 | 373 | 678 | 540 | 345 | 470 |
| phi-4 | 1,072 | 1,192 | 1,373 | 1,249 | 1,524 | 1,039 | 1,210 | 1,050 |
| Falcon3-10B-Instruct | 1,789 | 2,199 | 1,785 | 1,923 | 2,603 | 2,114 | 1,665 | 2,356 |

In this section, we present the results of *culture-specific* neurons for models not shown in Section 4.4. First, Table 16 shows the number of neurons identified by CULNIG-specific for each country. It is natural that the numbers are proportional to the total number of neurons (see Table 12) because the initial candidate neurons are the top 0.3% neurons ranked by attribution score. Subsequently, *culture-specific* neurons are refined by $CRC_{neur}$ and z-score, which may make the difference between countries. In the table, the number of neurons corresponding to South Korea tends to be large, indicating that models possess more dedicated neurons for South Korean culture than others.

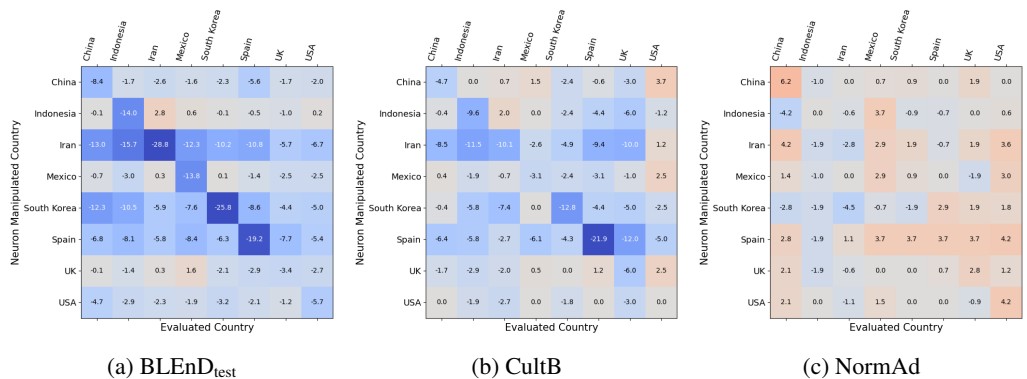

(a) BLEnD$_{test}$            (b) CultB            (c) NormAd

Figure 19: Score reductions after masking *culture-specific* neurons of gemma-3-27b-it.

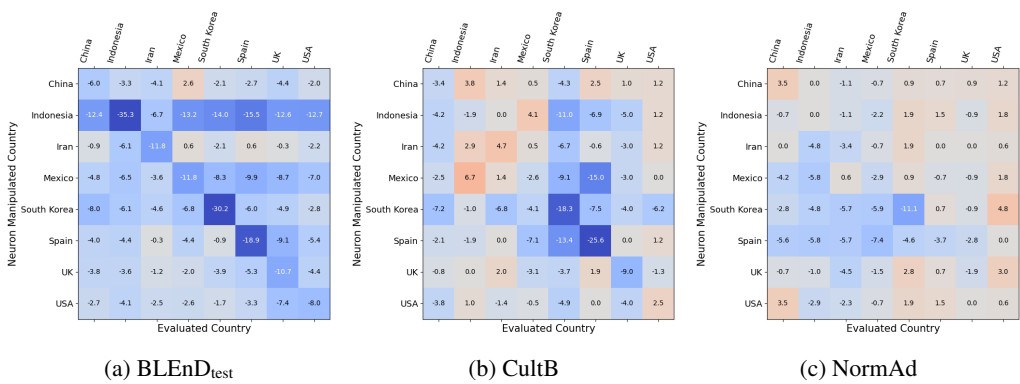

(a) BLEnD$_{test}$            (b) CultB            (c) NormAd

Figure 20: Score reductions after masking *culture-specific* neurons of Qwen3-14B.

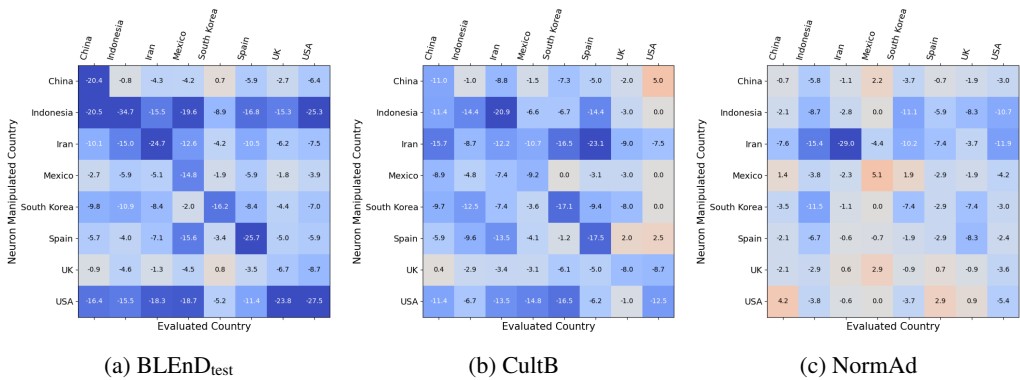

Figure 21: Score reductions after masking *culture-specific* neurons of Llama-3.1-8B-Instruct.

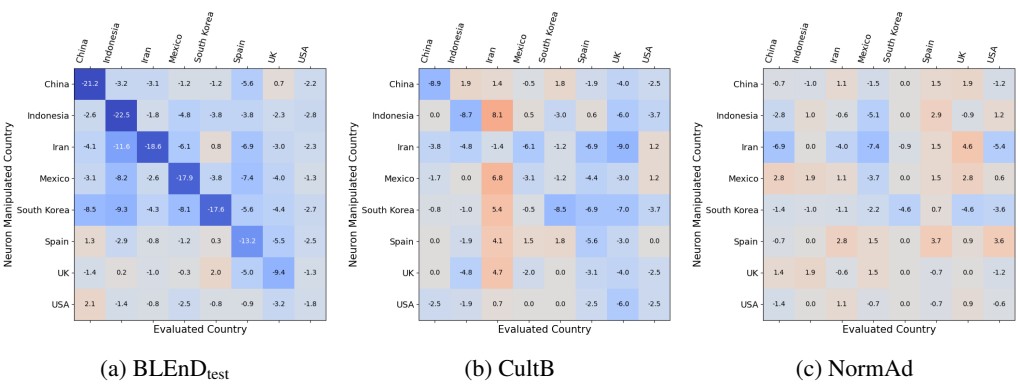

Figure 22: Score reductions after masking *culture-specific* neurons of phi-4.

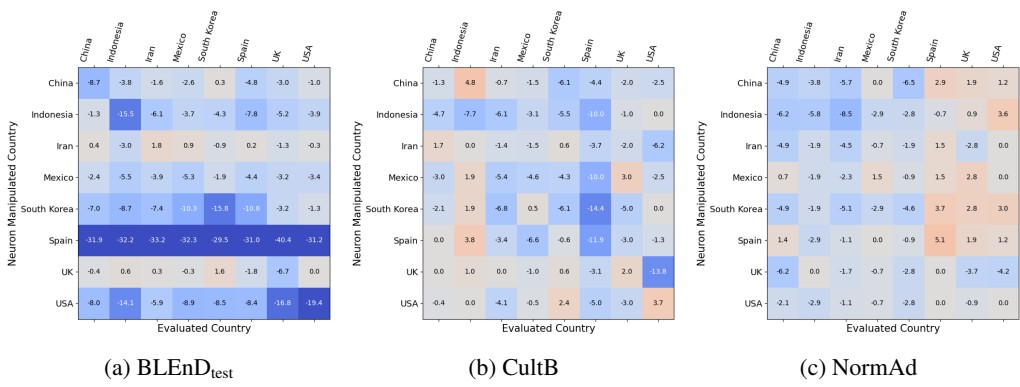

Figure 23: Score reductions after masking *culture-specific* neurons of Falcon3-10B-Instruct.

We show the evaluation results of *culture-specific* neurons for each model in Figure 4, Figure 19, Figure 20, Figure 21, Figure 22, and Figure 23. These figures show the reduction of scores compared to the original models for each problem culture. The result patterns are similar to Section 4.4. The scores of the same countries as the neuron targets are most affected for BLEnD$_{test}$ and CultB, while a clear pattern is not observed for NormAd. On the other hand, there are several cases where suppressing identified *culture-specific* neurons consistently degrades the scores for all evaluation cultures (e.g., Indonesian neurons in Llama-3.1-8B-Instruct on BLEnD$_{test}$). For these cases, the identified neurons may actually be important for understanding the benchmark task.

Table 17: Average ranks of performance drops among 16 cultures of BLEnD$_{test}$ when masking *culture-specific* neurons. Ranks are averaged over six models.

| Evaluation culture | Neuron culture | | | | | | | |
|---|---|---|---|---|---|---|---|---|
| | China | Indo-nesia | Iran | Mexico | South Korea | Spain | UK | USA |
| Algeria | 14.17 | 9.67 | 8.00 | 10.33 | 12.33 | 13.17 | 7.33 | 8.00 |
| Assam | 12.17 | 8.50 | 9.83 | 10.83 | 11.00 | 11.17 | 9.83 | 12.00 |
| Azerbaijan | 5.67 | 11.33 | 4.83 | 8.17 | 9.67 | 8.33 | 7.67 | 8.17 |
| China | **1.00** | 9.83 | 8.33 | 10.00 | 4.33 | 7.00 | 9.83 | 7.33 |
| Ethiopia | 13.50 | 12.50 | 12.83 | 11.50 | 12.33 | 14.00 | 14.33 | 12.50 |
| Greece | 8.33 | 10.67 | 8.50 | 12.50 | 8.33 | 8.17 | 7.67 | 8.17 |
| Indonesia | 8.83 | **1.17** | 4.00 | 4.00 | 5.33 | 4.67 | 8.00 | 6.33 |
| Iran | 6.33 | 11.50 | **3.50** | 10.00 | 9.50 | 8.50 | 11.67 | 8.83 |
| Mexico | 10.50 | 9.00 | 9.17 | **1.17** | 6.17 | 4.33 | 10.17 | 6.33 |
| Nigeria | 7.50 | 7.00 | 6.00 | 9.00 | 11.67 | 12.83 | 9.67 | 11.00 |
| North Korea | 7.50 | 11.33 | 13.00 | 11.67 | 6.50 | 12.83 | 13.00 | 12.50 |
| South Korea | 10.83 | 8.33 | 11.33 | 9.50 | **1.00** | 10.33 | 10.33 | 9.67 |
| Spain | 3.67 | 7.17 | 9.33 | 3.83 | 5.50 | **2.00** | 3.17 | 7.83 |
| UK | 7.83 | 8.17 | 10.83 | 8.67 | 9.67 | 3.17 | **1.17** | 5.67 |
| USA | 8.17 | 7.83 | 11.17 | 7.00 | 11.67 | 7.33 | 5.17 | **1.67** |
| West Java | 10.00 | 2.00 | 5.33 | 7.83 | 11.00 | 8.17 | 7.00 | 10.00 |

Table 18: In-class and out-of-class average rankings of score reduction when masking *culture-general* neurons in gemma-3-12b-it. Cultures are classified based on regions.

| Neuron culture | In-class | Out-of-class |
|---|---|---|
| China | **8.76** | 9.21 |
| Indonesia | **8.97** | 9.00 |
| Iran | **8.09** | 9.48 |
| Mexico | **7.00** | 9.13 |
| South Korea | **8.19** | 9.71 |
| Spain | **5.67** | 9.44 |
| UK | **5.42** | 9.54 |

Table 19: In-class and out-of-class average rankings of score reduction when masking *culture-general* neurons in gemma-3-12b-it. Cultures are classified based on spoken languages.

| Neuron culture | In-class | Out-of-class |
|---|---|---|
| Mexico | **3.83** | 9.36 |
| South Korea | **6.50** | 9.18 |
| Spain | **4.33** | 9.26 |
| UK | **7.42** | 9.23 |
| USA | **8.34** | 9.05 |

For a deeper analysis, we show the average rankings of performance drops among 16 cultures of BLEnD$_{test}$ when masking *culture-specific* neurons in Table 17. The ranks are averaged over six models, and a lower rank means a significant drop. We observe that the top ranks are always when the neuron target culture and evaluation culture agree, validating that the identified neurons especially contribute to their target culture.

Additionally, when *culture-specific* neurons of a specific culture are masked, it tends to have an impact on scores of related cultures. For example, when Mexican neurons are masked, Spain is the second most strongly influenced culture. When Spanish neurons are masked, Mexico is the third most influenced, and the second most influenced culture is the UK. Spain and Mexico are historically connected, and Spain and the UK are geographically close. In order to quantify these

cultural relationships, we classify the cultures included in BLEnD based on regions and spoken languages and compare in-class and out-of-class average rankings of score reduction when masking *culture-specific* neurons. For regions, we classify the cultures into Asia (Assam, Azerbaijan, China, Indonesia, Iran, North Korea, South Korea, West Java), Africa (Algeria, Ethiopia, Nigeria), Americas (Mexico, USA), and Europe (Greece, Spain, UK). For spoken languages, we choose some languages for analysis: English (Nigeria, UK, USA), Korean (North Korea, South Korea), and Spanish (Mexico, Spain). The results are shown in Table 18 and Table 19. The ranking of the neurons' own culture is excluded. In all cases, in-class cultures are affected more significantly than out-of-class cultures. These results confirm that *culture-specific* neurons contribute not only to their own culture but also to related cultures.

## G   REPLICATION OF LAPE AND CAPE

As discussed in Section 4.3 and Appendix D, the neuron distributions of CULNIG and CAPE are different. CAPE is a method designed to isolate culture neurons from language neurons of LAPE, using a multilingual and multicultural dataset, MUREL. We replicate the experiments of LAPE and CAPE according to their GitHub repository[17] and compare with our results.

LAPE identifies language-specific neurons using multilingual corpora taken from Wikipedia[18] (Wikimedia Foundation). Similarly, CAPE first selects neurons using MUREL (in this paper, we call this neuron set "MUREL neuron" to avoid conflict with the name "culture neuron" with CULNIG), and then refines neurons by excluding corresponding LAPE neurons to obtain "pure" culture neurons. As MUREL contains six languages and cultures (Danish (da), German (de), English (en), Persian (fa), Russian (ru), and Chinese (zh)), we identify neurons of these languages and cultures. For the model, we use gemma-3-12b-it.

Table 20: Neuron counts of LAPE and CAPE neurons in gemma-3-12b-it. Languages (cultures) are Danish (da), German (de), English (en), Persian (fa), Russian (ru), and Chinese (zh).

|  | da | de | en | fa | ru | zh |
|---|---|---|---|---|---|---|
| LAPE | 914 | 1,087 | 773 | 1,115 | 1,157 | 2,440 |
| MUREL | 412 | 477 | 1,059 | 718 | 810 | 3,897 |
| pure | 60 | 80 | 644 | 264 | 221 | 2,462 |

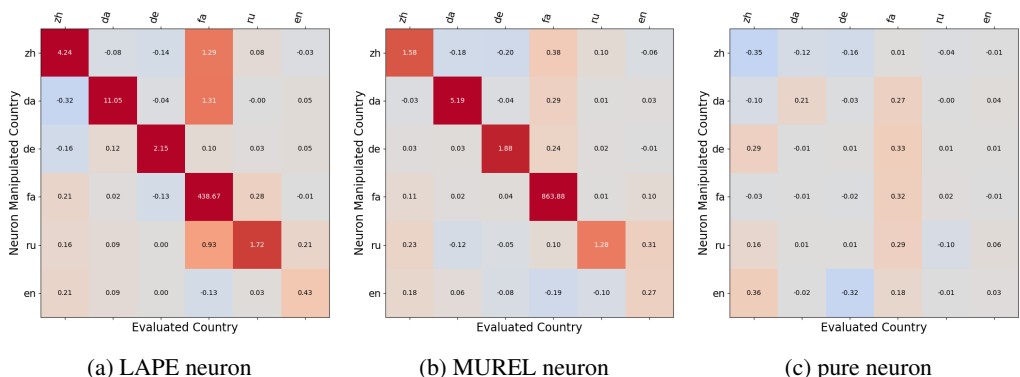

| (a) LAPE neuron | (b) MUREL neuron | (c) pure neuron |
|---|---|---|

Figure 24: Perplexity increase when masking LAPE, MUREL, and pure neurons from the original state of gemma-3-12b-it.

The number of identified neurons by LAPE and CAPE is shown in Table 20. The number of pure neurons is small, especially for da (60) and de (80). This indicates that the overlaps between LAPE and MUREL neurons are large, failing to isolate culture neurons from language neurons. For evaluation in the CAPE paper, they use the MUREL test set and see the perplexity change. We present the replicated evaluation results in Figure 24. It shows that for LAPE and MUREL neurons, increases in perplexity are most significant when the language or culture of neurons and data match. However, this is not the case for pure neurons, which have little impact after masking them. Based on these results, we speculate that most of the MUREL neurons are actually language neurons. Note that they use gemma-3-12b-**pt** in the original experiment, while we use gemma-3-12b-**it**, which is developed by performing instruction tuning on gemma-3-12b-pt, for consistency with our experiment. Other possible differences are hyperparameters, such as the context length of inputs.

Moreover, the evaluation results on BLEnD$_{test}$, CultB, and NormAd for LAPE, MUREL, and pure neurons are presented in Figure 25, Figure 26, and Figure 27, respectively. We show the score

---

[17]https://github.com/namazifard/Culture_Neurons/tree/
f48acc08d2d4a9117610f3e8e29a502fca2704c4

[18]https://huggingface.co/datasets/wikimedia/wikipedia

changes from the original model on problems of the cultures common in MUREL and each benchmark. As a result, none of the three methods caused significant changes to the scores. One plausible reason is that all the problems in these benchmarks are asked in English in our evaluation. As shown in Figure 24, the impacts on English are the smallest for all methods. Therefore, if identified neurons contribute to language abilities, not cultural understandings, the effects will be small when asking cultural questions in English.

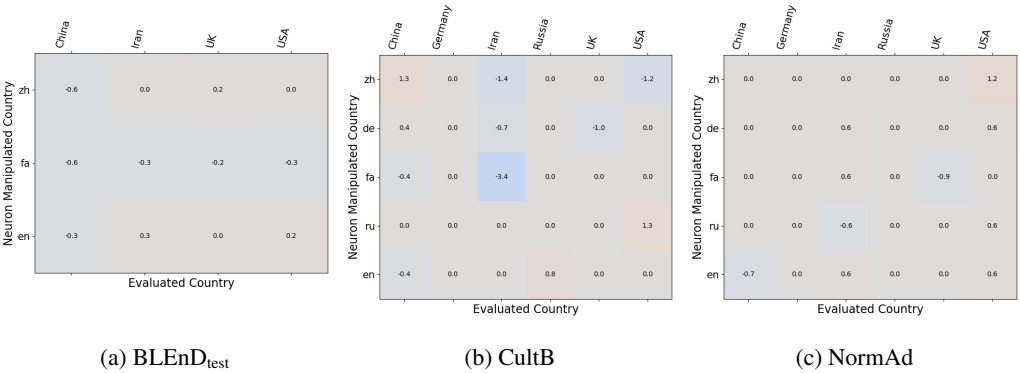

(a) BLEnD_test        (b) CultB        (c) NormAd

Figure 25: Score reductions after masking LAPE neurons of gemma-3-12b-it.

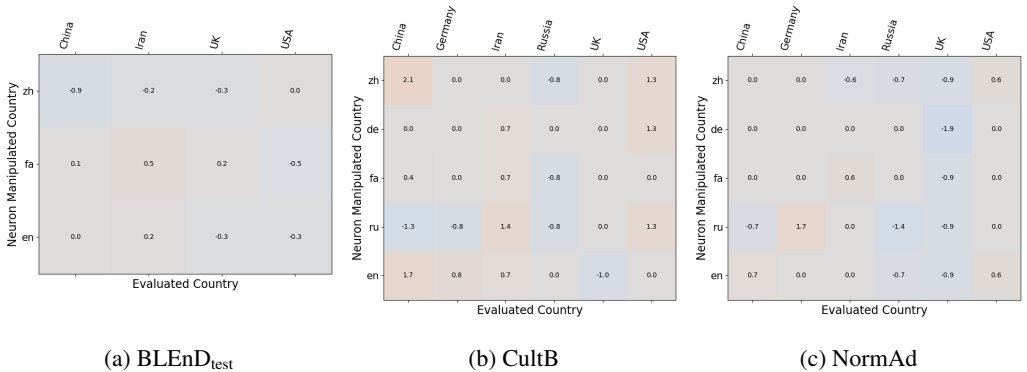

(a) BLEnD_test        (b) CultB        (c) NormAd

Figure 26: Score reductions after masking MUREL neurons of gemma-3-12b-it.

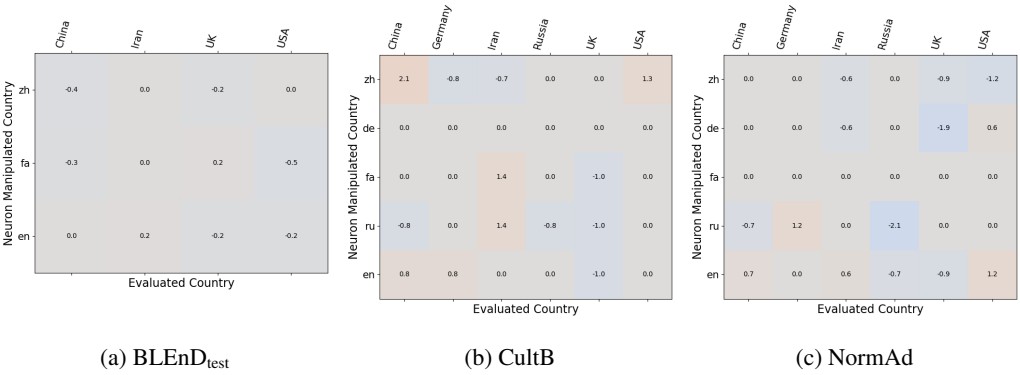

(a) BLEnD_test        (b) CultB        (c) NormAd

Figure 27: Score reductions after masking pure neurons of gemma-3-12b-it.

## H    INSTANCE-LEVEL NEURON ATTRIBUTION SCORES

As described in Section 4.5, we investigate the attribution scores of *culture-general* and *culture-specific* neurons in gemma-3-12b-it per instance in $\text{BLEnD}_{\text{neur}}$ to gain insight into what these neurons actually encode and compute. If neurons act as cultural-context gating signals to switch reasoning modes depending on the cultural background implied by the input, then they should have positive scores on most instances. If neurons encode category-specific cultural features, they should exhibit high scores on instances of a specific category.

We select the highest, middle, and lowest-scoring neurons from *culture-general* neurons in gemma-3-12b-it and plot the scores per instance in Figure 5, Figure 28, and Figure 29, respectively. We plot scores only for instances where the model assigns the probability $> 0.5$ to the correct token. The figures show that a *culture-general* neuron has positive scores on only $20 \sim 30\%$ of the instances and that the high-scoring instances span across categories and target countries. These results suggest that the neurons encode knowledge-level concepts rather than meta-level signals and that various kinds of concepts are encoded regardless of their types. Together with the results of Table 2, which show that masking *culture-general* neurons causes score reductions on all four cultural benchmarks but not on NLU benchmarks, it is indicated that *culture-general* neurons encode cultural concepts intensively, including knowledge and values, but do not contribute to universal knowledge or other NLU abilities.

Moreover, we plot the per-instance scores of *culture-specific* neurons in Figure 30. We observe that the top-scoring Iran-specific neuron has high scores on instances of Iran and Azerbaijan, and the top-scoring South Korea-specific neuron has high scores on instances of South Korea and North Korea. These are both neighboring countries, so *culture-specific* neurons can intensively encode information about their own and related cultures. Meanwhile, only $38\%$ of instances of Iran have positive scores for the top Iran-specific neuron, indicating that they do not encode meta-level signals.

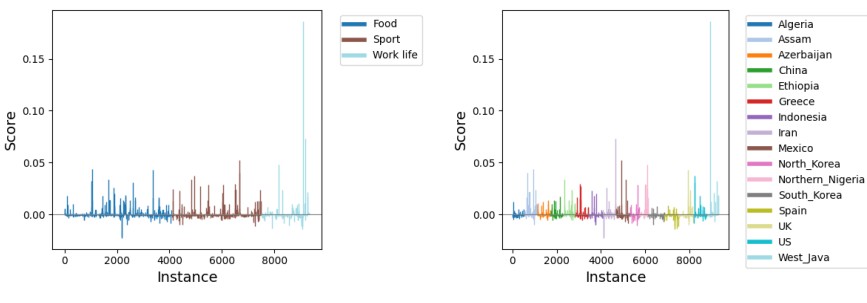

(a) Colored by question categories.     (b) Colored by question target cultures.

Figure 28: Attribution scores of the middle scoring *culture-general* neurons in gemma-3-12b-it ($3514^{th}$ neuron in MLP gate projection in the 15th layer) per instance in $\text{BLEnD}_{\text{neur}}$.

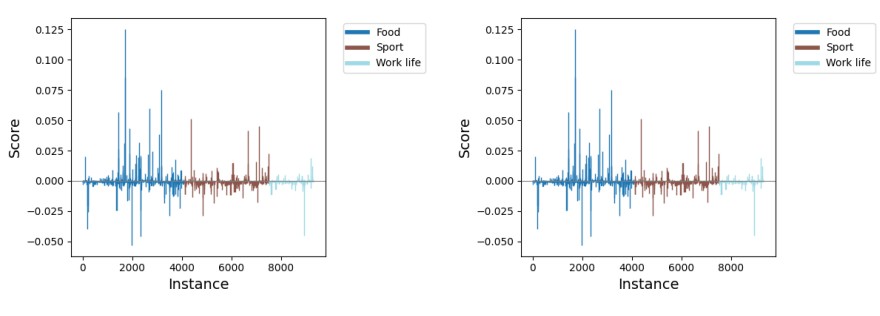

(a) Colored by question categories.     (b) Colored by question target cultures.

Figure 29: Attribution scores of the bottom scoring *culture-general* neurons in gemma-3-12b-it ($6850^{th}$ neuron in MLP gate projection in the 13th layer) per instance in $\text{BLEnD}_{\text{neur}}$.

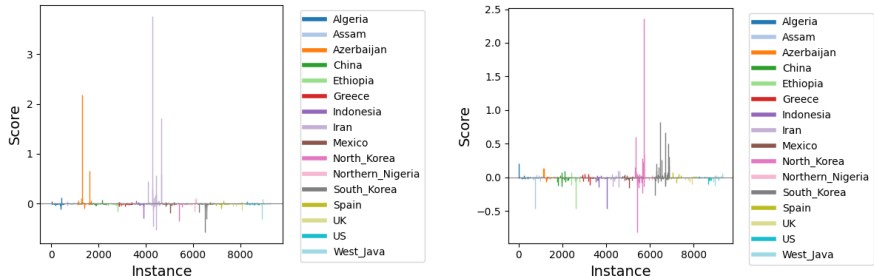

(a) The top-scoring Iran-specific neuron colored by cultures.

(b) The top-scoring South Korea-specific neuron colored by cultures.

Figure 30: Attribution scores of the top-scoring Iran-specific neuron ($3384^{th}$ neuron in MLP gate projection in the 8th layer) and South Korea-specific neuron ($12146^{th}$ neuron in MLP gate projection in the 13th layer) in gemma-3-12b-it per instance in BLEnD$_{neur}$.

# I   USING NORMAD TO IDENTIFY CULTURE-GENERAL NEURONS

The results in Section 4.3 show that masking *culture-general* neurons identified only by $\text{BLEnD}_{\text{neur}}$, a benchmark for everyday cultural knowledge, degrades the scores on all four cultural benchmarks. These results suggest that *culture-general* neurons capture broad representations of cultural understanding, including knowledge and values. In order to further analyze how these cultural aspects are encoded in neurons, we re-run CULNIG-general using NormAd as a cultural benchmark instead of BLEnD and identify *culture-general* neurons. Since the problems in NormAd have *yes*, *no*, or *neutral* labels as answer options, we split NormAd in a label-stratified manner, evenly dividing instances of each label, and then combine these halves to form $\text{NormAd}_{\text{neur}}$ and $\text{NormAd}_{\text{test}}$. In addition, we create $\text{NormAd}_{\text{ctrl}}$ by leaving a country name blank in Table 6.

As a result, we identify 7,989 neurons from gemma-3-12b-it, while the number is 8,087 when using $\text{BLEnD}_{\text{neur}}$. Let $N_{\text{BLEnD}}$ and $N_{\text{NormAd}}$ denote the neuron sets identified with $\text{BLEnD}_{\text{neur}}$ and $\text{NormAd}_{\text{neur}}$, respectively. The evaluation results when masking neurons are shown in Table 21.

Table 21: The evaluation results of gemma-3-12b-it when *culture-general* neurons identified using $\text{BLEnD}_{\text{neur}}$ or $\text{NormAd}_{\text{neur}}$ are masked.

| | #Neuron | BLEnD$_{\text{test}}$ | CultB | NormAd$_{\text{test}}$ | WVB | ComQA | QNLI | MRPC |
|---|---|---|---|---|---|---|---|---|
| (orig) | 0 | 64.22 | 78.08 | 55.42 | 64.08 | 79.71 | 75.37 | 78.04 |
| $N_{\text{BLEnD}}$ | 8,087 | 37.93 | 62.00 | 50.73 | 58.46 | 75.10 | 72.77 | 78.65 |
| $N_{\text{NormAd}}$ | 7,989 | 47.44 | 68.36 | 49.73 | 60.07 | 77.81 | 73.40 | 78.12 |
| $N_{\text{BLEnD}} \cap N_{\text{NormAd}}$ | 1,347 | 55.19 | 73.66 | 54.10 | 62.63 | 79.12 | 76.53 | 78.23 |
| $N_{\text{BLEnD}} \cup N_{\text{NormAd}}$ | 14,729 | 34.74 | 57.23 | 47.14 | 54.49 | 73.81 | 72.47 | 78.03 |
| $N_{\text{BLEnD}} \setminus N_{\text{NormAd}}$ | 6,740 | 51.39 | 68.87 | 51.09 | 62.30 | 76.15 | 72.52 | 78.72 |
| $N_{\text{NormAd}} \setminus N_{\text{BLEnD}}$ | 6,642 | 62.24 | 74.37 | 50.53 | 62.99 | 78.58 | 73.92 | 77.75 |

We observe that masking $N_{\text{NormAd}}$ degrades the scores on all cultural benchmarks, and so does masking $N_{\text{BLEnD}}$. Moreover, masking neurons in $N_{\text{BLEnD}} \setminus N_{\text{NormAd}}$ still harms $\text{NormAd}_{\text{test}}$ and WVB, suggesting that even if a neuron does not have high scores on $\text{NormAd}_{\text{neur}}$, it can contribute to $\text{NormAd}_{\text{test}}$ if it has high scores on $\text{BLEnD}_{\text{neur}}$. These results indicate that cultural knowledge and values can be encoded in the same neurons, rather than separable.

Next, $N_{\text{BLEnD}} \cup N_{\text{NormAd}}$ has a greater impact on the cultural benchmarks than $N_{\text{BLEnD}}$ or $N_{\text{BLEnD}}$ alone. This trend indicates that our pipeline, which only uses $\text{BLEnD}_{\text{neur}}$, may overlook some neurons that contribute to cultural understanding, although the score gaps are not that big between $N_{\text{BLEnD}}$ and $N_{\text{BLEnD}} \cup N_{\text{NormAd}}$.

We also observe that when we exclude neurons in $N_{\text{NormAd}} \setminus N_{\text{BLEnD}}$, they have a minimal impact on the cultural benchmarks except for $\text{NormAd}_{\text{test}}$. While $\text{BLEnD}_{\text{neur}}$ has 12,701 instances, $\text{NormAd}_{\text{neur}}$ consists of 5,048 instances, which focus on cultural etiquette. This narrow scope can lead to missed detection and noisy selection, so irrelevant neurons can be contained. Moreover, the score reductions caused by masking $N_{\text{BLEnD}}$ are not that different from those with $N_{\text{BLEnD}} \cup N_{\text{NormAd}}$, suggesting that $\text{BLEnD}_{\text{neur}}$ is more comprehensive than $\text{NormAd}_{\text{neur}}$. Thus, we believe that using $\text{BLEnD}_{\text{neur}}$ in CULNIG is better than $\text{NormAd}_{\text{neur}}$.

## J    DETAILS OF MODEL TRAINING

Table 22: Selection of top-culture and bottom-culture modules. Values in parentheses denote the number of culture neurons contained in each module.

| top-culture | bottom-culture |
|---|---|
| layer13 MLP (785) | layer0 attention (0) |
| layer12 MLP (685) | layer2 attention (0) |
| layer14 MLP (612) | layer8 attention (0) |
| layer10 MLP (491) | layer27 attention (0) |
| layer11 MLP (469) | layer28 attention (0) |
| layer7 MLP (429) | layer30 attention (0) |
| layer9 MLP (409) | layer31 attention (0) |
| | layer32 attention (0) |
| | layer36 attention (0) |
| | layer39 attention (0) |
| | layer41 attention (0) |
| | layer43 attention (0) |
| | layer44 attention (0) |
| | layer45 attention (0) |
| | layer47 attention (0) |
| | layer34 MLP (0) |
| | layer41 MLP (0) |
| | layer43 MLP (0) |

In this section, we present the supplementary information and results of Section 4.6. As described in Section 4.6, we select top-culture and bottom-culture modules to fine-tune gemma-3-12b-it with QNLI and MRPC. The selected modules are shown in Table 22. The top-culture modules are all MLP modules of shallow to middle layers, while the bottom-culture modules consist of the shallowest attention, deep attention, and deep MLP modules. This selection matches the distribution of culture-general neurons (subsection 4.3 and Appendix D). Note that for the bottom-culture modules, we randomly picked modules from those without any culture-general neurons to account for 10% of the total parameters. During training, we use AdamW (Loshchilov & Hutter, 2019) optimizer and linear scheduler with batch size 16. For QNLI, we randomly selected 10,000 training samples to reduce computational cost.

The evaluation results when the learning rate is 3e-5 are shown in Section 4.6 (Figure 6). We also show the results when the learning rate is 1e-5 and 5e-5 in Figure 31 and Figure 32, respectively. When the learning rate is 1e-5, there are almost no impacts on cultural benchmarks regardless of updated modules. When the learning rate is 5e-5, the scores on cultural benchmarks degrade at early steps when updating top-culture modules, and the scores also decrease for bottom-culture modules as the training goes on. Regarding the target benchmarks (QNLI or MRPC), the scores improve on all cases except for QNLI when targeting top-culture modules. These results suggest that forgetting can be avoided depending on the learning rate (or other parameters), but tuning bottom-culture modules can achieve a better outcome.

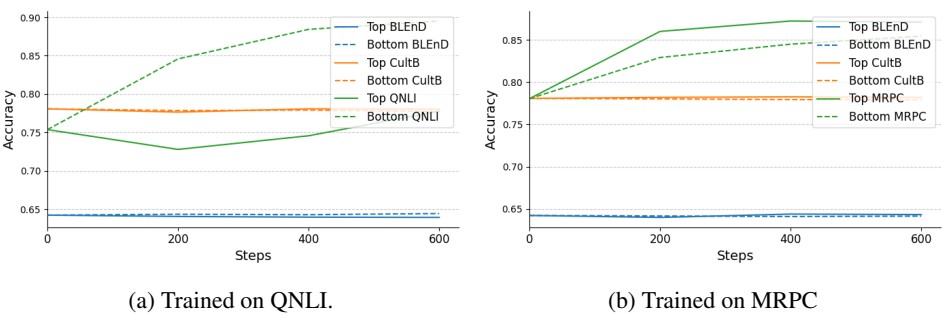

(a) Trained on QNLI.

(b) Trained on MRPC

Figure 31: Evaluation results when lr=1e-5.

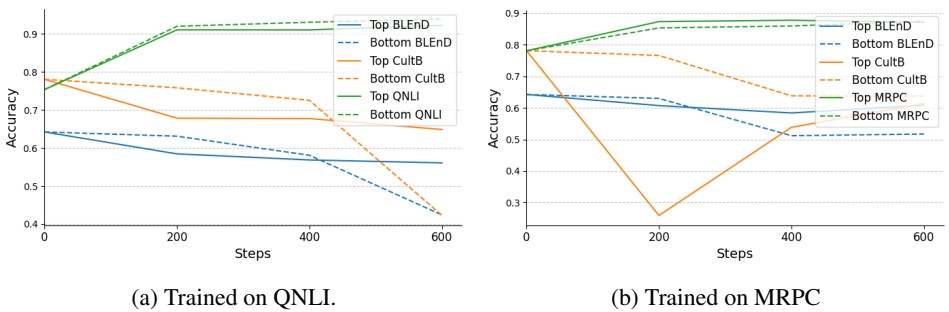

(a) Trained on QNLI.

(b) Trained on MRPC

Figure 32: Evaluation results when lr=5e-5.

# K    ABLATION OF DATASETS USED FOR NEURON IDENTIFICATION

Table 23: Evaluation results on $CRC_{test}$ when masking neurons identified with and without $CRC_{neur}$.

| Model | orig | w/ $CRC_{neur}$ | w/o $CRC_{neur}$ |
|---|---|---|---|
| gemma-3-12b-it | 100.00 | 100.00 | 99.75 |
| gemma-3-27b-it | 100.00 | 100.00 | 100.00 |
| Qwen3-14B | 100.00 | 100.00 | 100.00 |
| Llama-3.1-8B-Instruct | 100.00 | 96.00 | 0.00 |
| phi-4 | 100.00 | 100.00 | 0.13 |
| Falcon3-10B-Instruct | 100.00 | 99.62 | 98.25 |

CULNIG identifies culture neurons using $BLEnD_{neur}$, $BLEnD_{ctrl}$, and $CRC_{neur}$ (Section 3.3). In this section, we perform the ablation studies of these datasets. Table 23 compares the evaluation results on $CRC_{test}$ when masking neurons identified by CULNIG-general with and without $CRC_{test}$. We observe that without $CRC_{neur}$, masking identified neurons significantly reduces accuracy for some models. As described in Section 3.3, we use $CRC_{neur}$ in CULNIG to eliminate superficial neurons activated by tokens of country names, since such neurons should not be considered as supporting cultural mechanisms. The results confirm that $CRC_{neur}$ filters out such neurons.

Table 24: Evaluation results of gemma-3-12b-it when masking neurons identified by CULNIG-general with and without $BLEnD_{ctrl}$.

| | #Neuron | $BLEnD_{test}$ | CultB | NormAd | WVB | ComQA | QNLI | MRPC |
|---|---|---|---|---|---|---|---|---|
| orig | 0 | 64.22 | 78.08 | 58.54 | 64.08 | 79.71 | 75.37 | 78.04 |
| w/ ctrl | 8,087 | 37.93 | 62.00 | 52.02 | 58.46 | 75.10 | 72.77 | 78.65 |
| w/o ctrl | 6,494 | 39.65 | 61.57 | 52.82 | 62.28 | 70.13 | 67.49 | 78.25 |

Moreover, Table 24 shows the ablation results of $BLEnD_{ctrl}$ on gemma-3-12b-it. We observe that without $BLEnD_{ctrl}$, the evaluation scores on the NLU benchmarks are worse than normal CULNIG-general, although the number of neurons is smaller. This result indicates that neurons that contribute to properties other than cultural understanding, such as language understanding, tend to get high scores and be selected without $BLEnD_{ctrl}$. These results confirm that the datasets used in our pipeline are important for accurately and steadily identifying culture neurons.

## L    COMPARISON OF NEURON ATTRIBUTION SCORES

As explained in Section 3.2, we adopt a gradient-based score to measure neuron attribution in solving cultural problems, following Yang et al. (2024). In this section, we compare it with alternative attribution methods.

In our method, the attribution score of a neuron at the $i$-th token position is calculated as Equation 1, and aggregated across tokens by taking the maximum (Equation 2). As alternatives, we consider:

- Mean aggregation (*mean*): replacing the maximum with the mean across token positions.

- Weight-gradient inner product (*norm*): directly computing inner product $\boldsymbol{w} \cdot \frac{\partial P(y|x)}{\partial \boldsymbol{w}}$ for the subkey $\boldsymbol{w}$ (row vectors of MLP gate, attention query, key, and value modules) associated with each neuron.

Table 25: Evaluation results of masking *culture-general* neurons identified with *max* (the one used in the original pipeline), *mean*, and *norm* attribution scores on gemma-3-12b-it.

| Score | #Neuron | BLEnD$_{\text{test}}$ | CultB | NormAd | WVB | ComQA | QNLI | MRPC |
|---|---|---|---|---|---|---|---|---|
| (orig) | 0 | 64.22 | 78.08 | 58.54 | 64.08 | 79.71 | 75.37 | 78.04 |
| *max* | 8,087 | 37.93 | 62.00 | 52.02 | 58.46 | 75.10 | 72.77 | 78.65 |
| *mean* | 8,151 | 59.50 | 75.75 | 58.59 | 64.27 | 79.20 | 74.22 | 78.23 |
| *norm* | 8,151 | 56.54 | 75.06 | 58.86 | 63/72 | 79.71 | 74.86 | 78/20 |

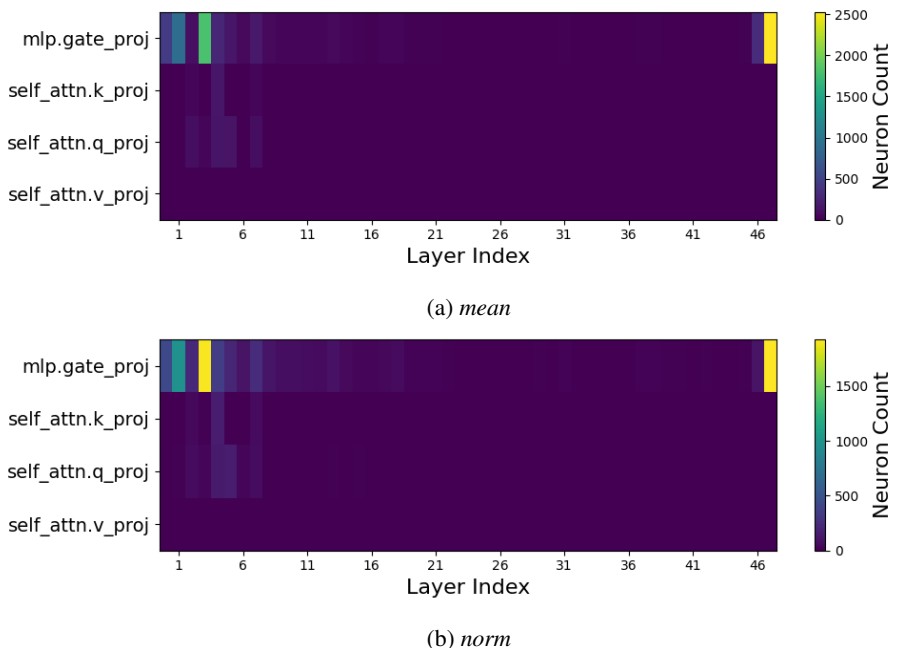

(a) *mean*

(b) *norm*

Figure 33: The distribution of neurons identified with *mean* and *norm* attribution scores.

We identify *culture-general* neurons in gemma-3-12b-it with *mean* and *norm* scores integrated into CULNIG-general. The evaluation results are shown in Table 25. Masking *mean* or *norm* barely affects the benchmark scores, indicating that identified neurons do not engage in model behavior. Figure 33 shows that such neurons are mainly located in very shallow and very deep MLP layers, unlike the distribution of our original method (Figure 3). Moreover, Figure 34 compares the distribution of attribution scores. For *max*, the distribution has a wider positive tail, while for *mean* and *norm*, only a few neurons have a positive score. Actually, the number of neurons with z-score $\geq 2.5$ is 729 for *max*, but only 6 for *mean* and 15 for *norm*, suggesting that *mean* and *norm* failed to

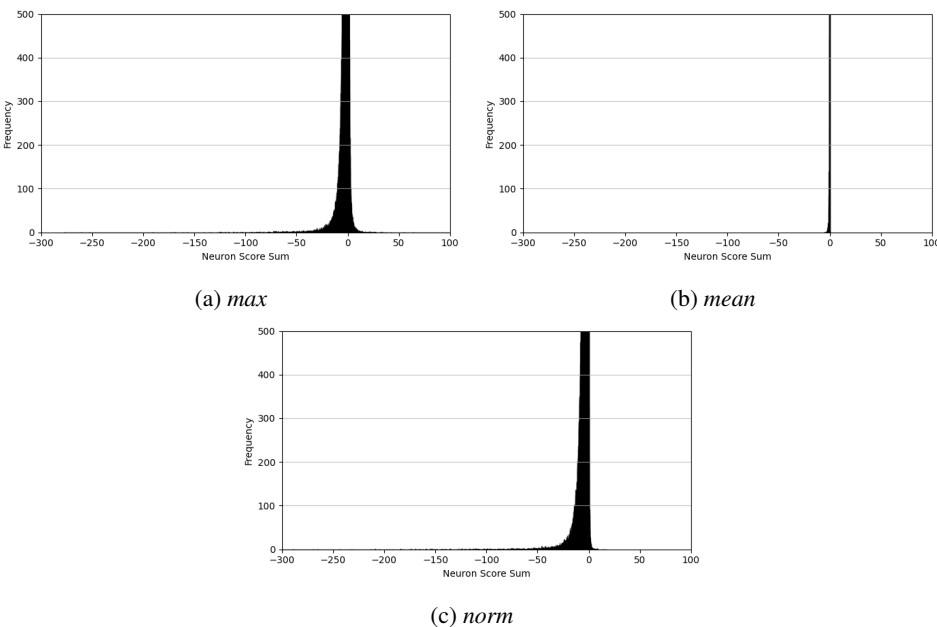

(a) *max*

(b) *mean*

(c) *norm*

Figure 34: The distribution of neuron attribution scores with *max*, *mean*, and *norm* on gemma-3-12b-it.

distinguish neurons that contribute to cultural understanding. We speculate that this is because the scores of *mean* and *norm* take into account all token positions. Not all tokens necessarily encode cultural representations, so attribution can be obscure. In contrast, *max* highlights salient tokens, which may result in the best performance for identifying culture neurons.

## M  EXPERIMENTAL CONFIGURATION

In our experiments, we used NVIDIA H100 GPUs. To calculate all neuron attribution scores on $BLEnD_{neur}$, $BLEnD_{ctrl}$ and $CRC_{neur}$ in CULNIG, it took up to 4 hours per model with one H100 GPU (for gemma-3-27b-it, we used two H100 GPUs). For fine-tuning in Section 4.6, it took up to 20 minutes to train gemma-3-12b-it for 600 steps.

## N  LLM USAGE

We utilized ChatGPT to construct the CRC dataset and to generate task instructions in the evaluation prompts. We also used ChatGPT and Gemini[19] to proofread the paper. When implementing the scripts for our experiments, we used GitHub Copilot[20] as a coding assistant.

---

[19]https://gemini.google/about/
[20]https://github.com/features/copilot

