# OpenReview forum: "Neuron-Level Analysis of Cultural Understanding in Large Language Models"
_ICLR.cc/2026/Conference — ICLR 2026 Poster_

### Official Review · Reviewer_MVRH · 2025-10-23

**Soundness:** 3
**Presentation:** 3
**Contribution:** 2
**Rating:** 4
**Confidence:** 4

**Summary:**

The paper introduces CULNIG, a gradient-based pipeline for identifying neurons that support cultural understanding in large language models. By leveraging a BLEnD control dataset and a CountryRC filter, CULNIG isolates “culture-general” and “culture-specific” neurons while excluding those merely responsive to country names or task formats. The authors find that less than 1% of neurons, mainly in shallow-to-mid MLP layers, are responsible for most culture-related behavior, and masking them causes a 30% performance drop on cultural benchmarks with minimal NLU impact. They further show that culture-specific neurons affect historically or linguistically related cultures and that fine-tuning culture-rich modules can erode cultural ability, highlighting implications for reliable model adaptation.

**Strengths:**

(1) This paper employs dual control datasets (BLEnDctrl, CRC) to eliminate spurious lexical or task-format activations.

(2) This paper provides clear empirical evidence that <1% of neurons drive most cultural behavior. And reveals layer-level localization of cultural knowledge in shallow–mid MLPs.

(3) Demonstrates cross-model and cross-benchmark robustness across multiple open LLMs.

(4) Code and data with a detailed README file are shared.

**Weaknesses:**

(1) The paper’s definition of “cultural understanding” is narrow and tied to country-labeled QA tasks, which may not capture deeper cultural reasoning or values.

(2) This paper conflates cultural knowledge with cultural alignment (norms, values, empathy), yet reports all under a single “culture-general” construct.

(3) The paper does not clearly articulate why gradient-based neuron analysis is superior to attention- or concept-level analysis for cultural representation.

(4) A key limitation is that the application study only shows a negative case—fine-tuning culture-rich modules harms cultural competence—without demonstrating how the identified neurons can be leveraged to enhance or transfer cultural understanding.

**Questions:**

(1) Can the identified culture-related neurons be leveraged to improve or transfer cultural understanding, rather than only showing degradation when masked or fine-tuned?

(2) How sensitive are the results to the chosen thresholds (percentiles, z-scores, filtering ratios)?

(3) How do you distinguish between neurons representing knowledge (facts about culture) versus values or norms? Could a separate analysis clarify this conceptual distinction?

(4) Prompt robustness: You mention using multiple prompt formats—what is the variance in neuron identification and benchmark scores across these prompts?

(5) For culture-specific neurons, do related cultures (e.g., neighboring or linguistically similar) consistently show correlated effects? Could you quantify this relationship?

---

> ### Author Response · Authors · 2025-11-20
> **Response to Reviewer MVRH (1/2)**
>
> We appreciate your constructive and thoughtful feedback and thank you for taking the time to read our code and data. We summarize our response below.
>
> > **Weakness 1 and 2** The definition of cultural understanding is narrow and conflated
>
> We explain our definition of cultural understanding in the Introduction, lines 71-73. The scope of cultural understanding is (a) knowledge specific to particular cultures and (b) the ability to capture differences in values across cultural backgrounds. According to this definition, we selected cultural benchmarks: BLEnD and CulturalBench for knowledge, NormAd as a task involving both knowledge and values, and WVB for values. Using these benchmarks, we comprehensively evaluate the cultural understanding of LLMs, including both knowledge and values, and our _culture-general_ neurons capture both aspects.
>
> > **Weakness 3** Why we chose gradient-based neuron analysis for cultural understanding
>
> As described in Section 2.2, we use a gradient-based method to quantify the genuine contribution of neurons to final outputs. Compared to activation-based methods, gradient-based methods can consider negative activation values. Additionally, since a cultural context is not necessarily encoded in every token in a sentence, we thought that activation probabilities are not suitable for identifying culture neurons. Moreover, since a neuron is a unit component of LLMs, a neuron-level analysis can investigate the mechanisms of LLMs with finer granularity than attention head or concept-level analysis.
>
> > **Weakness 4 and Question 1** How to improve cultural understanding of LLMs based on the findings
>
> First, to the best of our knowledge, no previous studies have devised methods to identify culture neurons other than CAPE. Our main contribution and scope are to find such neurons and analyze the mechanisms underlying the cultural understanding of LLMs.
>
> While we demonstrate that updating modules that contain many _culture-general_ neurons can harm cultural understanding of LLMs when fine-tuning with QNLI/MRPC, we believe that our findings can be leveraged to improve cultural understanding of LLMs. For example, when we employ knowledge editing methods (e.g., [1, 2]) to update existing cultural knowledge of LLMs, it would be better to target early/mid MLP modules. In addition, when we insert new cultural knowledge into models, we can selectively update early/mid MLP neurons that are not included in _culture-general_ neurons so that existing knowledge will not be disrupted. We do not include complete propositions and experiments in our paper, but we will add this discussion as future work.
>
> > **Question 2** The sensitivity of our results to the thresholds
>
> We conducted sensitivity analyses in the additional experiment 3 in General Response. Please refer to it.
>
> > **Question 3** How to distinguish neurons representing cultural knowledge and values?
>
> In the additional experiment 2 in General Response, we re-ran CULNIG using NormAd. We conclude that cultural knowledge and values can be encoded in the same neurons. For the details, please refer to General Response.

---

> > ### Author Response · Authors · 2025-11-20
> > **Response to Reviewer MVRH (2/2)**
> >
> > > **Question 4** The variance in neuron identification and benchmark scores across these prompts when using different prompt variations
> >
> > We performed the additional analysis regarding this point.
> > In CULNIG, we use four prompt instructions for CRC to calculate attribution scores and aggregate scores over all prompts. We compare the top 1% of neurons in gemma-3-12b-it when using only the first prompt versus only the second prompt. As a result, out of 7,372 MLP and 3,932 attention neurons, the number of overlapping neurons is 2,544 and 2,036, respectively. The common neurons are selected to some extent, but as $\text{CRC}\_{\text{neur}}$ has only 200 instances, some variance is observed. Therefore, we consider that it is better to use multiple prompts to mitigate the noise.
> >
> > Next, we calculate the standard deviations of evaluation scores of gemma-3-12b-it across four prompt variations. We exclude $\text{BLEnD}\_{\text{test}}$ because instructions are already included in the problems of BLEnD, and we did not prepare instructions.
> >
> > ||avg score|std|
> > |--|--|--|
> > |CulturalBench|78.08|0.5839|
> > |NormAd|58.54|0.7586|
> > |WVB|64.08|0.9916|
> >
> > Although the standard deviations are not large, we consider that using multiple prompts would make the evaluation precise and reliable.
> >
> > > **Question 5** Could we quantify the relationships between related cultures when masking _culture-specific_ neurons?
> >
> > We conducted further analysis regarding this point. We classify the cultures included in $\text{BLEnD}\_{\text{test}}$ based on regions and spoken languages, and calculate the in-class and out-of-class average rankings of score reduction when masking _culture-specific_ neurons.
> >
> > ### Region
> > We classify cultures into the following regions:
> >
> > - Asia: China, North Korea, South Korea, Indonesia, West Java, Iran, Assam, Azerbaijan
> > - Africa: Algeria, Nigeria, Ethiopia
> > - Americas: USA, Mexico
> > - Europe: Greece, Spain, UK
> >
> > The mean rankings of score reduction when masking _culture-specific_ neurons are shown below. The ranking of _culture-specific_ neurons’ own culture is excluded.
> >
> > |neuron culture|in-region|out-of-region|
> > |--|--|--|
> > |China|8.76|9.21|
> > |Indonesia|8.97|9.0|
> > |Iran|8.09|9.48|
> > |Mexico|7.0|9.13|
> > |South Korea|8.19|9.71|
> > |Spain|5.67|9.44|
> > |UK|5.42|9.54|
> > |USA|6.33|9.14|
> >
> > In every case, in-region average rankings are smaller than out-of-region averages, indicating that _culture-specific_ neurons have a greater impact on their own region’s culture.
> >
> > ### Spoken Language
> > We classify cultures according to their spoken languages and picked some groups for analysis:
> >
> > - English: USA, UK, Nigeria
> > - Spanish: Spain, Mexico
> > - Korean: North Korea, South Korea
> >
> > The mean rankings of score reduction when masking _culture-specific_ neurons are shown below. The ranking of _culture-specific_ neurons’ own culture is excluded.
> >
> > |neuron culture|in-class|out-of-class|
> > |--|--|--|
> > |Mexico|3.83|9.36|
> > |South Korea|6.5|9.18|
> > |Spain|4.33|9.26|
> > |UK|7.42|9.23|
> > |USA|8.34|9.05|
> >
> > Again, we observe that masking _culture-specific_ neurons has a greater effect on the cultures where the same language is spoken. Overall, we quantitatively confirm the relationships between the related cultures in terms of _culture-specific_ neurons.
> >
> > [1] Meng et al. ICLR 2023. Mass-Editing Memory in a Transformer
> > [2] Fang et al. ICLR 2025. AlphaEdit: Null-Space Constrained Knowledge Editing for Language Models

---

> > > ### Comment · Reviewer_MVRH · 2025-11-26
> > >
> > > Thank the authors for the additional clarification. However, some concerns are not fully addressed:
> > >
> > > (1) While your rebuttal clarifies how each benchmark corresponds to ‘knowledge’ or ‘values,’ the operationalization of cultural understanding remains narrow. The construct is still effectively reduced to performance on country-labeled QA/classification tasks, which capture only shallow cultural cues rather than deeper reasoning, norms, or perspective-taking. I recommend reframing claims more modestly to reflect that the analysis pertains to these specific tasks and benchmarks, not to cultural understanding in a broader cognitive sense.
> > >
> > > (2) The paper still only demonstrates a negative application. The response outlines speculative future directions but does not provide any empirical evidence showing that the identified neurons can improve, enhance, or transfer cultural understanding. Since this was one of the core questions motivating the original concern, the limitation remains: the work identifies impactful neurons but does not yet show how these findings can be leveraged constructively.

---

> > > > ### Author Response · Authors · 2025-11-27
> > > > **Response to Reviewer MVRH**
> > > >
> > > > We appreciate the reviewer’s continuous feedback. We agree that our argument is not sufficient. Below is our response.
> > > >
> > > > > (1) Operationalization of cultural understanding remains narrow
> > > >
> > > > As you point out, we evaluate _culture-general_ neurons using benchmarks on cultural knowledge, etiquette, and values, which may overlook a broader cognitive sense, such as perspective-taking. Although we define the scope of cultural understanding in our paper as two dimensions, (a) knowledge specific to particular cultures and (b) the ability
> > > > to capture differences in values across cultural backgrounds, it may not be clear enough. Meanwhile, our evaluation is not limited to QA or classification tasks. We evaluate models on WarldValuesBench, a task to predict survey responses about participants’ values based on demographic attributes, which requires reasoning grounded in cultural values. Considering these points, we will clarify the scope of cultural understanding that our evaluation takes into account and acknowledge that our settings may miss some cognitive aspects in the Limitations.
> > > >
> > > > > (2) Future work on improving cultural understanding of LLMs
> > > >
> > > > Our claim that our findings can be leveraged to enhance or transfer cultural understanding of LLMs is based on empirical evidence from previous studies and plausible hypotheses. First, several studies proposed methods to identify language-specific neurons [1, 2]. They show that the ability to process a specific language is largely dependent on a small number of neurons, which is similar to our _culture-general_ and _culture-specific_ neurons. Based on these findings, Zhao et al. [3] proposed a method to selectively train language-specific neurons, thereby enhancing the multilingual abilities of LLMs. This situation is comparable to our analysis. Moreover, we demonstrated that related cultures tend to share _culture-specific_ neurons in Section 4.4 and Appendix F. This nature indicates that improving understanding of one culture may transfer to its related cultures. Based on these evidences, we claim that our findings can be utilized for improving or transferring cultural understanding of LLMs. We will add this discussion to Section 4.6.
> > > >
> > > > Once again, we are really grateful for your constructive comments. We believe that our study has improved greatly during this rebuttal period. We would like to hear if you have any concerns left. Thanks!
> > > >
> > > > [1] Tang et al. ACL 2024. Language-Specific Neurons: The Key to Multilingual Capabilities in Large Language Models
> > > > [2] Kojima et al. NAACL 2024. On the Multilingual Ability of Decoder-based Pre-trained Language Models: Finding and Controlling Language-Specific Neurons
> > > > [3] Zhao et al. NeurIPS 2024. How do Large Language Models Handle Multilingualism?

---

### Official Review · Reviewer_7fEB · 2025-10-26

**Soundness:** 3
**Presentation:** 3
**Contribution:** 3
**Rating:** 6
**Confidence:** 3

**Summary:**

This paper investigates the internal mechanisms underlying cultural understanding in large language models (LLMs). While previous studies have documented cultural bias or disparities in model outputs, little is known about how LLMs internally represent and process cultural information. To address this, the authors propose CULNIG (Cultural Neuron Identification Pipeline with Gradient-based Scoring)— a neuron-level interpretability framework that uses gradient attribution combined with control datasets to identify neurons responsible for cultural reasoning.
CULNIG distinguishes between culture-general neurons and culture-specific neurons. Through extensive experiments on six open-source LLMs, the authors show that these neurons constitute less than 1% of all neurons, are concentrated in shallow-to-middle MLP layers, and that masking them leads to substantial degradation (up to 30%) on cultural benchmarks such as BLEnD, CulturalBench, NormAd, and WorldValuesBench—while general NLU benchmarks remain largely unaffected.
The paper also presents an engineering case study showing that fine-tuning modules rich in culture-general neurons causes a loss of cultural understanding, suggesting a practical pathway for parameter-efficient and culturally robust fine-tuning.

**Strengths:**

The study tackles an important and underexplored question: how LLMs internally encode culture. Given the growing concerns around fairness and cross-cultural bias in LLMs, this direction is highly relevant to the ICLR community.

CULNIG combines gradient-based attribution with two control datasets to precisely isolate neurons genuinely related to cultural inference. This design elegantly avoids the pitfalls of activation-based methods, which often conflate linguistic or topical cues with cultural representations.

The evaluation spans multiple models, benchmarks, and modalities. The consistency of results across architectures strongly supports the validity of the findings. The experiments are well-documented and reproducibility is addressed.

The finding that culture-related neurons cluster in mid-level MLP layers aligns with prior “knowledge neuron” research, suggesting that cultural reasoning may rely on factual recall mechanisms rather than contextual inference. The observed cross-cultural interference (e.g., Mexico–Spain) provides additional interpretive depth.

**Weaknesses:**

CULNIG relies primarily on BLEnD, which emphasizes everyday cultural knowledge (e.g., food, work, sports). Although the authors claim generalization to higher-level cultural attributes such as values or etiquette (NormAd, WVB), Section 4.4 admits that the pattern is “less clear” for NormAd. This suggests that the current pipeline is more effective at identifying *cultural knowledge neurons* than *value-oriented or normative neurons*, and that the captured concept of “culture” may be biased toward knowledge representation rather than deeper socio-cognitive dimensions.

Key thresholds in CULNIG are selected heuristically based on inflection points in Figure 2. The main conclusion—that fewer than 1% of neurons govern cultural understanding—could depend on these particular values. A sensitivity analysis (e.g., varying thresholds from 0.5% to 2%) would strengthen claims about sparsity and robustness.

The reported layer distribution (shallow-to-middle) contradicts CAPE, which located culture neurons mainly in upper layers. Although the authors hypothesize methodological causes (gradient attribution vs. activation probability, accuracy vs. perplexity metrics), the inconsistency remains unclarified, leaving open a significant question for the field.

The paper defines these neurons functionally—as those beneficial across cultural tasks—but does not explore *what they actually compute*. Do they encode an abstract notion of “cultural context,” or simply a meta-cognitive control signal for out-of-context reasoning? A deeper probe into their semantic or computational roles would improve interpretability.

**Questions:**

1. CULNIG mainly relies on BLEnD${\mathrm{neur}}$ (a knowledge-oriented dataset) for neuron identification. As observed in Section 4.4, culture-specific neurons show weaker patterns on NormAd (etiquette/value tasks). Does this suggest that “cultural knowledge” and “cultural values” are encoded by different neurons or mechanisms? If CULNIG were re-run using NormAd or WVB as $D{\mathrm{neur}}$, would you expect to discover a distinct set of neurons?
2. The main claims—such as $ <1% $ sparsity and the clear separation between cultural and NLU functionality—depend on the thresholds $ t_{\text{MLP}} = 1% $ and $ t_{\text{attn}} = 0.2% $ selected in Section 4.2. How stable are these findings under threshold variation? For example, if $ t_{\text{MLP}} $ increases to $ 2% $ or $ 3% $, would NLU tasks (e.g., QNLI) start to degrade? Such analysis would clarify the boundary between cultural and general-purpose neurons.
3. These neurons generalize across tasks and attributes, but beyond their functional importance, do you have any hypotheses about what they compute? Do they encode high-level “cultural semantics” or act as meta-reasoning units that signal when external knowledge is needed? Clarifying this would help bridge the gap between functional discovery and mechanistic interpretation.

---

> ### Author Response · Authors · 2025-11-20
> **Response to Reviewer 7fEB**
>
> We would like to thank the reviewer for acknowledging the importance of investigating the cultural mechanisms of LLMs and for their valuable feedback on our experimental design. Our response is as follows:
>
> > **Weakness 1 and Question 2** Do identified neurons mainly encode cultural knowledge and less cultural values? Are cultural knowledge and cultural values encoded in different neurons?
>
> In the additional experiment 2 in General Response, we re-run CULNIG using NormAd instead of BLEnD to identify _culture-general_ neurons. Please refer to it.
>
> > **Weakness 4 and Question 3** Deeper analysis into what the identified neurons encode
>
> Please check the additional experiment 1 in General Response.
>
> > **Weakness 2 and Question 2** Sensitivity analysis to the thresholds
>
> We conducted the sensitivity analysis in the additional experiment 3 in General Response. Please refer to it.
>
> > **Weakness 3** The results contradict those of CAPE
>
> As we state in Section 4.4 and show in Figure 11 (Figure 12 in the revised version), our culture neurons are concentrated in early/mid MLP layers, while CAPE neurons are mainly in the very first and last layers. The distribution of CAPE neurons resembles that of language-specific neurons [1]. Recent studies have demonstrated that LLMs process multilingual prompts through three stages: (1) map multilingual inputs into the shared representation, (2) process semantic information in the shared space, and (3) translate back for generation [2, 3]. Based on these observations, we can hypothesize that CAPE neurons are primarily responsible for processing token patterns specific to the target culture. This hypothesis aligns with the difference of the attribution scores (gradient-based or activation-based) and metrics (QA accuracy or perplexity). We will add this discussion to Section 4.4.
>
> [1] Tang et al. ACL 2024. Language-Specific Neurons: The Key to Multilingual Capabilities in Large Language Models
> [2] Wang et al. ACL 2025. Lost in Multilinguality: Dissecting Cross-lingual Factual Inconsistency in Transformer Language Models
> [3] Wu et al. ICLR 2025. The Semantic Hub Hypothesis: Language Models Share Semantic Representations Across Languages and Modalities

---

> > ### Comment · Reviewer_7fEB · 2025-11-25
> >
> > Thank you for your clarification. I believe most of my concerns have been addressed.

---

### Official Review · Reviewer_tC8F · 2025-10-28

**Soundness:** 3
**Presentation:** 3
**Contribution:** 3
**Rating:** 6
**Confidence:** 3

**Summary:**

This paper uses a gradient-based attribution metric to identify neurons relevant for models’ performance on global/cultural understanding benchmarks. The authors are able to find both cultural-general neurons and cultural-specific ones. They also show that modular fine-tuning of LMs informed by locations of culture-relevant neurons can avoid diminishing LMs’ cultural understanding.

**Strengths:**

Overall, this paper is well-written and organized in a logical manner.

During several points in the paper, I saw that there was some care involved in the authors’ experimental decisions. For example, to show that their findings around culture-related neurons generalize, the authors split their multiple choice question dataset into train-like and test sets based on topical categories (e.g. if a neuron pertains to culture, its presence in food questions should generalize to its presence in holiday questions). As additional examples of care, the authors also experiment with multiple prompt variations and also account for cases where neurons may pertain to surface level features rather than culture, e.g. country names.

I appreciate that the authors dedicate a section of their paper to applications of their findings. Often, a pitfall of mechanistic interpretability research is that though its findings may satisfy one’s curiosities around the inner workings of language models, the practical implications of such findings are not always demonstrated.

**Weaknesses:**

This paper focuses on identifying culture neurons, but the paper doesn’t really give us much insight that justifies this conceptual focus. The paper reuses existing methods from Yang et al. 2024 to identify neurons, and its application of its findings (module section for fine-tuning) also doesn’t seem to need to be culture-specific. What makes identifying and working with culture neurons unique from identifying other types of neurons? It’s good that it seems like you managed to find them, but what does your work offer us that identifying any other kind of topical neuron doesn’t offer?

The authors’ last sentence of their conclusion also hints at some of my qualms: “While our findings do not directly improve the cultural understanding of LLMs, they provide a foundation for future studies to do so.” More evidence of this foundation and its implications specifically for culture would make this paper stronger.

**Questions:**

I wasn’t quite sure why there was so much space dedicated to a discussion of how transformer neurons work in 3.1?

---

> ### Author Response · Authors · 2025-11-20
> **Response to Reviewer tC8F**
>
> We appreciate your recognition of our effort in designing an accurate pipeline and evaluation settings, as well as their practical application. Below are our responses.
>
> > **Weakness 1** What makes identifying and working with culture neurons unique from identifying other types of neurons?
>
> First, understanding the mechanisms behind cultural understanding of LLMs is important because it can promote the development of fair and inclusive LLMs. It has not been investigated well, so we focus on this point. Moreover, to achieve this goal, we adjusted the method of Yang et al. (2024), rather than simply reusing it. We used $\text{BLEnD}\_{\text{ctrl}}$ to exclude neurons related to task understanding and CRC to remove neurons that respond to county names, which are specific procedures in cultural neuron isolation. However, we believe that our pipeline design can be leveraged to identify neurons related to other topics as well.
>
> > **Weakness 2** How to improve cultural understanding of LLMs based on the findings
>
> To the best of our knowledge, no previous studies have identified culture neurons other than CAPE, so our main contribution and scope are to find such neurons and analyze the mechanisms of cultural understanding of LLMs. In addition, we show in Section 4.5 (4.6 in the revised version) that when fine-tuning models with QNLI or MRPC, updating modules that contain many _culture-general_ neurons can compromise the cultural understanding of LLMs. This finding gives us an interpretable explanation of the side effects of LLM training, which will lead to practical applications aimed at preventing such degradation.
>
> Furthermore, we believe that our results can be leveraged to improve cultural understanding of LLMs. For example, when we want to update existing cultural knowledge of LLMs, we can employ knowledge editing methods (e.g., [1, 2]). Here, we can target early/mid MLPs because cultural knowledge is mainly encoded in those modules. In addition, when we insert new cultural knowledge, we can update early/mid MLP neurons that are not included in _culture-general_ neurons so that new knowledge is easily incorporated while retaining existing knowledge. We do not include complete designs and experiments in our paper, but we will add this discussion as future work.
>
> This claim is based on empirical evidence from previous studies and plausible hypotheses. First, in the case of language neurons, several studies [3, 4] show that the multilingual abilities of LLMs are largely dependent on a small subset of neurons. Based on these findings, Zhao et al. [5] proposed a method to selectively train language-specific neurons, thereby enhancing the multilingual abilities of LLMs. This situation is comparable to our analysis. Moreover, we demonstrated that related cultures tend to share culture-specific neurons in Section 4.4 and Appendix F. This nature indicates that improving understanding of one culture may transfer to its related cultures. Based on these evidences, we claim that our findings can be utilized for improving or transferring cultural understanding of LLMs. We will add this discussion to Section 4.6 in the revised version.
>
> > **Question 1** Why is so much space dedicated to a discussion of how transformer neurons work in 3.1?
>
> We agree that we can make this section shorter to accommodate the new experiments conducted during the discussion period. We will revise it.
>
> [1] Meng et al. ICLR 2023. Mass-Editing Memory in a Transformer
> [2] Fang et al. ICLR 2025. AlphaEdit: Null-Space Constrained Knowledge Editing for Language Models
> [3] Tang et al. ACL 2024. Language-Specific Neurons: The Key to Multilingual Capabilities in Large Language Models
> [4] Kojima et al. NAACL 2024. On the Multilingual Ability of Decoder-based Pre-trained Language Models: Finding and Controlling Language-Specific Neurons
> [5] Zhao et al. NeurIPS 2024. How do Large Language Models Handle Multilingualism?

---

### Official Review · Reviewer_cBSz · 2025-10-31

**Soundness:** 3
**Presentation:** 3
**Contribution:** 3
**Rating:** 6
**Confidence:** 4

**Summary:**

The paper investigates how cultural understanding is represented at the neuron level in LLMs. It introduces CULNIG (CULture Neuron Identification with Gradient-based Scoring), a pipeline for identifying neurons that contribute to cultural understanding. Using gradient-based attribution and multiple control datasets, the authors isolate Culture-general neurons and Culture-specific neurons, and show that masking them reduces performance on cultural benchmarks but not on general NLU benchmarks, masking culture-specific neurons affect both the target and related cultures on cultural benchmarks, and fine-tuning modules containing many culture-general neurons on general NLU tasks reduces cultural understanding.

**Strengths:**

1. First systematic neuron-level investigation of cultural mechanisms in LLMs, extending prior interpretability work (bias, language, factual neurons)
2. CULNIG uses a rigorous multi-step filtering pipeline (BLEnD neur – ctrl – CRC control) to remove spurious correlations
3. Generalization of result on six modern open source models and across benchmarks
4. Practical insight demonstrating the effect of fine-tuning on cultural competence depending on target modules

**Weaknesses:**

1. Ambiguous scope of "cultural understanding": all benchmarks have different coverage of cultures and definition of cultural values/knowledge, the authors should clarify on this work's scope of cultures and definitions of cultural understanding.
2. Limited interpretability: the paper identifies neuron clusters but does not probe what these neurons encode semantically (e.g., cultural values vs. lexical associations).
3. Fine-tuning only tests QNLI/MRPC, so it's unclear how results scale to other downstream tasks or general instruction tuning.
4. Country-token filtering (CRC) may not fully remove lexical signals and cultural context could still leak through token patterns, so the authors should have a discussion/ablation/evaluation on this.

**Questions:**

1. If masking culture-general neurons doesn’t affect NLU, does that imply they’re unused for NLU or because of neuron and layer redundancy?
2. Definition of “cultural understanding” needs to be clearer. The paper mixes factual, normative, and value-based culture; boundaries between these types are not operationally clean.
3. Prompting procedure – The role of ChatGPT-generated prompts for CRC/BLEnD not clearly justified (possible data leakage or stylistic bias).
4. Authors claim generalization “to multilingual settings,” but examples and quantitative evidence are minimal.
5. Concentration in shallow/mid layers aligns with knowledge neurons, but it’s not clear whether this overlaps with existing factual memory neurons.

---

> ### Author Response · Authors · 2025-11-20
> **Response to Reviewer cBSz**
>
> We appreciate the constructive comments and questions from the reviewer. Below are our responses.
>
> > **Weakness 1 and Question 2**: Definition of cultural understanding
>
> In our paper, we have already defined cultural understanding in the Introduction, lines 71-73. The scope of cultural understanding is (a) knowledge specific to particular cultures and (b) the ability
> to capture differences in values across cultural backgrounds. We selected cultural benchmarks according to this definition: BLEnD and CulturalBench for knowledge, NormAd as a task involving both knowledge and values, and WVB for values.
>
> > **Weakness 2**: Deeper analysis into what the identified neurons encode
>
> Please refer to the additional experiment 1 in General Response.
>
> > **Weakness 3**: Scalability of fine-tuning experiments to other tasks and settings
>
> There are numerous downstream tasks and settings. QNLI and MRPC are well-established and widely used benchmarks for measuring the general natural language understanding of LLMs, so we chose them in our experiments. Our findings in Section 4.5 (4.6 in the revised paper version) can be applied to other tasks as well. As you point out, it is true that our experiments are limited to these benchmarks, so we will include this point in the Limitations.
>
> > **Weakness 4**: CRC may not be sufficient to remove all lexical signals
>
> We employ CRC to remove superficial neurons that simply respond to country names. As you indicate, it does not necessarily remove all the lexical signals. We consider it almost impossible to deal with every pattern of lexical signals, since it is difficult to list such patterns without any leakage. Meanwhile, previous studies that isolate topical neurons usually do not consider the lexical signals. In our method, we address the evident pattern of country names, as cultural questions always contain country names in their sentences.
>
> We conducted the ablation study of CRC in Appendix H (Appendix K in the revised version). Table 18 (Table 23 in the revised version) shows that when CRC is not used in CULNIG, the evaluation results become unstable when masking identified neurons. This result validates the effectiveness of using CRC in the pipeline.
>
> > **Question 1**: Are _culture-general_ neurons unused for NLU?
>
> Since masking _culture-general_ neurons does not affect the scores on NLU benchmarks (Table 2), we can consider that these neurons are not used for NLU. Since masking them degrades scores on cultural benchmarks, it is indicated that cultural and NLU tasks are performed with a different set of neurons.
>
> > **Question 3**: Justification of ChatGPT-generated prompts for CRC/BLEnD
>
> First, we do not use ChatGPT-generated prompts for BLEnD, as the prompts of BLEnD (available on HuggingFace) already include the instructions. For CRC, we utilize ChatGPT to create the dataset. The purpose of CRC is to filter out superficial neurons that respond to country names, so it has the following requirements: (i) country names have to be recognized to answer the question, and (ii) it does not require cultural knowledge to answer the question. After the generation, we manually checked whether each instance meets these criteria. Additionally, when designing the CRC, we considered the variety of styles and made sure that 40% of instances contain two country names (one for the answer and one for the dummy).
>
> > **Question 4**: Evidence on generalization of _culture-general_ neurons to multilingual settings
>
> We conducted multilingual evaluations using short QAs of $\text{BLEnD}_{\text{test}}$, which is detailed in Appendix D (E in the revised version). We investigated six models across 13 languages and 16 cultures, and showed that in most cases, suppressing _culture-general_ neurons reduced accuracy more significantly than suppressing random neurons. We believe that these results provide evidence that the _culture-general_ neurons contribute even in multilingual settings.
>
> > **Question 5**: The overlaps between _culture-general_ neurons and existing studied knowledge neurons
>
> Table 2 indicates that the overlaps between _culture-general_ neurons and general commonsense knowledge neurons are small, since masking _culture-general_ neurons has only a minor impact on CommonsenseQA. Cultural and general commonsense knowledge may be encoded in a different set of neurons.

---

### Author Response · Authors · 2025-11-20
**General Response (1/3)**

We sincerely thank the reviewers for their insightful and valuable feedback. We are encouraged by the positive comments recognizing our paper’s contribution to elucidating mechanisms for cultural understanding of LLMs. In response to weaknesses and questions, we have conducted several additional experiments.

## 1. Deeper analysis into what _culture-general_ and _culture-specific_ neurons encode
Several reviewers note that our method identifies _culture-general_ and _culture-specific_ neurons, but we do not explore what these neurons actually compute and encode (question 2 by Reviewer cBSz, weakness 4, and question 3 by Reviewer 7fEB, question 3 by Reviewer MVRH).

To understand this, we investigate instance-level scores of neurons. We sample several neurons from _culture-general_ and _culture-specific_ neurons of gemma-3-12b-it and plot their attribution scores (computed as equation (7) in Section 3.2) per instance in $\text{BLEnD}_{\text{neur}}$. We plot scores only for instances where the model answers correctly with probability>0.5 (9325/12701 instances). If a neuron encodes meta-level control signals for cultural concepts or an abstract concept of culture, then it should have high scores on all the instances.

The results show that **a _culture-general_ neuron has positive scores on only 20~30% of the instances and that the high-scoring instances span across categories and target countries**. It suggests that these neurons encode knowledge-level concepts rather than meta-level signals and that various kinds of concepts are encoded regardless of their categories or origin. Together with Table 2, which shows that masking _culture-general_ neurons causes score reductions on all cultural benchmarks but not on NLU benchmarks, it is indicated that _culture-general_ neurons encode cultural concepts intensively, including knowledge and values, but not general commonsense knowledge or other NLU abilities. We can interpret that cultural information is concentrated in these neurons and separated from other abilities, but countries and categories are not organized.

On the other hand, _culture-specific_ neurons tend to exhibit high scores specifically on instances of their own and related culture. For example, the top Iran-specific neuron exhibits high scores on instances of Iran and Azerbaijan. Meanwhile, only 38% of Iranian instances have positive scores for that neuron. Therefore, _culture-specific_ neurons may encode knowledge of specific cultures concentratedly, but not at the meta-level.
We will add the figures and discussion to our paper as Section 4.5 and Appendix H in the revised version.

---

> ### Author Response · Authors · 2025-11-20
> **General Response (2/3)**
>
> ## 2. Using NormAd instead of BLEnD as a cultural benchmark in CULNIG
>
> Weakness 1 and question 1 by Reviewer 7fEB and question 3 by Reviewer MVRH ask whether cultural knowledge and values are encoded in different neurons or not. To answer this question, we re-run CULNIG on gemma-3-12b-it using NormAd as a cultural benchmark instead of BLEnD. Since the problems of NormAd have yes/no/neutral labels as answer options, we split NormAd in a label\-stratified manner, evenly dividing instances of each label, and then combine these halves to form $\text{NormAd}\_{\text{neur}}$ and $\text{NormAd}\_{\text{test}}$.
>
> As a result, 7,989 _culture-general_ neurons are identified (while 8,087 when using $\text{BLEnD}\_{\text{neur}}$), and the number of overlapping neurons between neurons identified with $\text{NormAd}\_{\text{neur}}$ and $\text{BLEnD}\_{\text{neur}}$ is 1,347. Moreover, we present the evaluation results when masking these neurons below.
>
> ||#neuron|$\text{BLEnD}\_{\text{test}}$|CulturalBench|$\text{NormAd}\_{\text{test}}$|WVB|ComQA|QNLI|MRPC|
> |--|--|--|--|--|--|--|--|--|
> |original|0|64.22|78.08|55.42|64.08|79.71|75.37|78.04|
> |$\text{BLEnD}\_{\text{neur}}$|8087|37.93|62.00|50.73|58.46|75.10|72.77|78.65|
> |$\text{NormAd}\_{\text{neur}}$|7989|47.44|68.36|49.73|60.07|77.81|73.40|78.12|
> |Intersection|1347|55.19|73.66|54.10|62.63|79.12|76.53|78.23|
> |Union|14729|34.74|57.23|47.14|54.49|73.81|72.47|78.03|
> |$\text{BLEnD}\_{\text{neur}}$ \ $\text{NormAd}\_{\text{neur}}$|6740|51.39|68.87|51.09|62.30|76.15|72.52|78.72|
> |$\text{NormAd}\_{\text{neur}}$ \ $\text{BLEnD}\_{\text{neur}}$|6642|62.24|74.37|50.53|62.99|78.58|73.92|77.75|
>
> We observe that masking neurons identified with $\text{NormAd}\_{\text{neur}}$ degrades the scores on all cultural benchmarks, and so does masking neurons with $\text{BLEnD}\_{\text{neur}}$. Moreover, when we exclude $\text{NormAd}\_{\text{neur}}$ neurons from $\text{BLEnD}\_{\text{neur}}$ neurons ($\text{BLEnD}\_{\text{neur}}$ \ $\text{NormAd}\_{\text{neur}}$), masking the neurons still harms $\text{NormAd}\_{\text{test}}$ and WVB. These results indicate that cultural knowledge and values can be encoded in the same neurons, rather than being separable. It aligns with the instance-level analysis in the additional experiment 1.
>
> Next, the union of neurons with $\text{BLEnD}\_{\text{neur}}$ and $\text{NormAd}\_{\text{neur}}$ has a greater impact on the cultural benchmarks than neurons identified with either $\text{BLEnD}\_{\text{neur}}$ or $\text{NormAd}\_{\text{neur}}$ alone. This trend indicates that our pipeline, which only uses $\text{BLEnD}\_{\text{neur}}$, may overlook some neurons that contribute to cultural understanding. Although the score gaps are not that big between $\text{BLEnD}\_{\text{neur}}$ and Union, we will add this point as a limitation.
>
> Meanwhile, we also observe that when we exclude $\text{BLEnD}\_{\text{neur}}$ neurons from $\text{NormAd}\_{\text{neur}}$ neurons ($\text{NormAd}\_{\text{neur}}$ \ $\text{BLEnD}\_{\text{neur}}$), they have a minimal impact on the cultural benchmarks except for $\text{NormAd}\_{\text{test}}$. While $\text{BLEnD}\_{\text{neur}}$ has 12,701 instances, $\text{NormAd}\_{\text{neur}}$ consists of 5,048 instances, which focus on cultural etiquette. This narrow scope can lead to the missed detection and the noisy selection, so irrelevant neurons can be contained. Moreover, the score reductions brought by masking neurons with $\text{BLEnD}\_{\text{neur}}$ are not that different from those with Union, suggesting that $\text{BLEnD}\_{\text{neur}}$ is more comprehensive than $\text{NormAd}\_{\text{neur}}$. Thus, we believe that using $\text{BLEnD}\_{\text{neur}}$ is better in CULNIG. The figures and discussions will be included in Section 4.5 and Appendix I in the revised version.

---

> ### Author Response · Authors · 2025-11-20
> **General Response (3/3)**
>
> ## 3. Sensitivity analysis to the thresholds for the proportion of selected neurons as _culture-general_ neurons
> Weakness 2 and question 2 by Reviewer 7fEB and question 2 by Reviewer MVRH ask about the sensitivity of the results to the thresholds, $t_{\text{MLP}}$ and $t_{\text{attn}}$. As Reviewer 7fEB points out, our claim that the cultural understanding of LLMs is driven by a sparse set of neurons accounting for less than 1% of the total depends on these thresholds. We determine the thresholds according to the results of Section 4.2; the evaluation score transitions as the thresholds increase. To strengthen our claim, we increase the thresholds $t_{\text{MLP}}$ and $t_{\text{attn}}$ and show the whole results on gemma-3-12b-it below.
>
> **$t_{\text{MLP}}$**
> |thresh|#neuron|$\text{BLEnD}_{\text{test}}$|CulturalBench|$\text{NormAd}_{\text{test}}$|WVB|ComQA|QNLI|MRPC|
> |--|--|--|--|--|--|--|--|--|
> |original|0|64.22|78.08|55.42|64.08|79.71|75.37|78.04|
> |1%|8,087|37.93|62.00|50.73|58.46|75.10|72.77|78.65|
> |2%|15,426|36.41|57.50|51.06|60.28|72.03|68.68|78.88|
> |3%|22,770|35.29|55.54|49.36|58.88|69.16|65.23|78.58|
>
> **$t_{\text{attn}}$**
> |thresh|#neuron|$\text{BLEnD}_{\text{test}}$|CulturalBench|$\text{NormAd}_{\text{test}}$|WVB|ComQA|QNLI|MRPC|
> |--|--|--|--|--|--|--|--|--|
> |original|0|64.22|78.08|55.42|64.08|79.71|75.37|78.04|
> |0.2%|8,087|37.93|62.00|50.73|58.46|75.10|72.77|78.65|
> |0.4%|8,868|37.84|61.80|51.46|58.71|75.20|73.40|79.03|
> |0.6%|9.652|37.50|61.04|51.60|58.56|74.88|73.42|79.19|
>
> When increasing $t_{\text{MLP}}$ to 2 or 3%, while the cultural scores decrease, the scores on ComQA and QNLI degrade equally or more significantly. Furthermore, increasing $t_{\text{attn}}$ to 0.4 or 0.6% has only a small effect on the scores. These results suggest that the neurons added when increasing the thresholds do not play a specific role in the cultural domain, which aligns with the results presented in Figure 2.
>
> Note that our goal is not to find the optimal thresholds in terms of scores. Through these analyses, we demonstrate that the LLMs’ cultural understanding primarily relies on sparse _culture-general_ neurons. We will include these results in Appendix C in the revised version of the paper.

---

> > ### Author Response · Authors · 2025-11-21
> > **Paper Revision Uploaded**
> >
> > We thank the reviewers again for their suggestions on the revisions to improve our paper. In response, we uploaded the revised version of the paper. We have highlighted the changes in the manuscript. The list of revisions is as follows:
> >
> > - **Instance-level analysis**: Section 4.5 and Appendix H
> > - **Running CULNIG with NormAd**: Section 4.5 and Appendix I
> > - **Sensitivity analysis to the thresholds**: Section 4.2 and Appendix C
> > - Remove redundant explanations in Section 4.2
> > - Add discussions about how to improve cultural understanding of LLMs based on our findings to Section 4.6
> > - Provide further discussion regarding the difference between our method and CAPE in Section 4.4
> > - Quantify the relationships of related cultures in terms of _culture-specific_ neurons in Appendix F

---

### Meta-Review · Area_Chair_DP1o · 2025-12-28

**Summary:**

This paper presents a thorough neuron-level investigation of cultural understanding in large language models via the proposed CULNIG pipeline, identifying sparse culture-general and culture-specific neurons that causally support performance on a range of cultural benchmarks while remaining largely orthogonal to general NLU abilities. Reviewers initially raised concerns regarding the scope and definition of “cultural understanding,” the interpretability of the identified neurons, sensitivity to heuristic thresholds, reliance on knowledge-oriented benchmarks, and the lack of constructive applications beyond degradation analyses. In response, the authors conducted substantial additional experiments, including instance-level probing of neuron behavior, re-running the pipeline with value-oriented benchmarks, comprehensive threshold sensitivity analyses, prompt-robustness checks, and quantitative analyses of cross-cultural relationships. These additions significantly strengthened the empirical grounding of the paper and clarified the intended scope and claims.

**Reviewer Concerns:**

Most major concerns were convincingly addressed in the rebuttal and revision. Questions about what culture-general and culture-specific neurons encode were answered through detailed instance-level attribution analyses, supporting the interpretation that these neurons capture distributed cultural knowledge and values rather than meta-level control signals. Concerns about whether cultural knowledge and values are separable were addressed by re-running CULNIG with NormAd, showing substantial overlap and indicating co-encoding. Sensitivity to threshold choices was carefully analyzed, demonstrating that the sparsity and cultural/NLU separation claims are robust. Methodological issues regarding CRC, prompt design, multilingual generalization, and overlaps with existing knowledge neurons were clarified with additional ablations and experiments.

**Reviewer Scores:**

All reviewers initially scored the paper around the acceptance threshold, with ratings ranging from marginally below to marginally above accept. Based on the scope and quality of the rebuttal, Reviewer 7fEB explicitly indicated that most concerns were addressed and would likely maintain or slightly increase their score. Reviewers cBSz and tC8F, whose concerns centered on interpretability, scope, and positioning, would likely view the added analyses and clarifications favorably and converge on a clear accept recommendation. Although Reviewer MVRH retained some reservations about breadth and constructive applications, the authors’ revisions and moderated claims likely raise the perceived contribution from borderline to acceptable.

---

### Decision · Program_Chairs · 2026-01-26

Accept (Poster)